# Rethinking Entropy Regularization in Large Reasoning Models

## Abstract

Reinforcement learning with verifiable rewards (RLVR) has shown great promise in enhancing the reasoning abilities of large reasoning models (LRMs). However, it suffers from a critical issue: entropy collapse and premature convergence. Naive entropy regularization, a common approach for encouraging exploration in the traditional RL literature, fails to address this problem in the context of LRM. Our analysis reveals that this failure stems from the vast action space and long trajectories in LRMs, which easily trigger a global entropy explosion as the model indiscriminately explores all possible actions and states. To address this, we propose **SIREN** (**S**elect**I**ve ent**R**opy r**E**gularizatio**N**), a method that confines exploration to a meaningful subset of actions and states. **SIREN** achieves this through a two-step entropy masking mechanism, consisting of a top-p mask and a peak-entropy mask. In addition, regularization is transformed into a self-anchored form to stabilize training. Across five mathematical benchmarks, **SIREN** attains superior average performance over previous entropy-related RLVR approaches, exemplified by a +6.6 maj@k improvement on AIME24/25 with Qwen2.5-Math-7B. Further analysis confirms that **SIREN** promotes greater response diversity and maintains entropy at an appropriate level, which helps to preserve the validation pass@k throughout training. This effectively mitigates the premature convergence problem common in RLVR for LRMs.

## 1 Introduction

Reinforcement learning with verifier rewards (RLVR) has become a promising method for enhancing large reasoning models (LRMs) by integrating reward modeling with automated verification. This technique has been shown to substantially improve complex reasoning abilities in domains such as mathematics and code (DeepSeek-AI, 2025; Team et al., 2025; Yang et al., 2025). Despite its success, a critical limitation has emerged in practice: entropy collapse and premature convergence (Yu et al., 2025; Cui et al., 2025). This issue causes the policy to become deterministic at an early stage, leading responses to collapse into near-identical outputs, and ultimately reducing both training efficiency in later stages and overall performance.

A common and intuitive solution to mitigate this issue is to incorporate entropy regularization to encourage exploration (Ziebart, 2010; Haarnoja et al., 2018b;c). In the context of LRMs, naive entropy regularization exhibits strong hyperparameter sensitivity and limited efficiency: a small coefficient yields minimal performance gains, while a large coefficient triggers a rapid entropy explosion(Cui et al., 2025). To better understand this behavior, we conduct a preliminary experiment. By analyzing probability distributions and the generated tokens from both an original model and one that experienced an entropy explosion under strong entropy regularization, we observe two key phenomena: (1) In the entropy-exploded model, the probability distribution becomes nearly uniform over the whole vocabulary, whereas in the original model, probability mass concentrates on a small, semantically meaningful subset. (2) In the entropy-exploded model, entropy remains uniformly high across almost all token positions, whereas in the original model, only a small fraction of tokens exhibit high entropy. Taken together, these phenomena suggest that the vast action space of LRMs makes it easy to increase the entropy by diffusing the probability mass across many low-utility tokens. The autoregressive generation process then propagates this uncertainty forward, causing the accumulated entropy bonus to scale with sequence length and ultimately trigger an explosion.

Based on these findings, we propose **S**elect**I**ve ent**R**opy r**E**gularizatio**N** (*SIREN*), a novel method that restricts exploration to a more effective subset of actions and states. *SIREN* consists of a two-step masking mechanism to identify exploration targets precisely: (1) a *top-p mask*, which limits exploration to semantically meaningful subsets within the large action space of LRMs; and (2) a *peak-entropy mask*, which limits exploration on several logically critical tokens along long trajectories. Furthermore, to calibrate the regularization strength and stabilize training, we reformulate the naive regularization as *self-anchored regularization*, which helps control the overall entropy level during training.

To evaluate *SIREN*'s performance, we conduct a comprehensive evaluation across multiple backbones and mathematical reasoning benchmarks. Our method achieves significantly improved performance on Qwen2.5-Math-7B, with a maj@k of **54.6** and an avg@k of **46.1**. Compared to other baselines that also incorporate entropy, *SIREN* improves the maj@k by **+4.8** and the avg@k by **+1.6**. *SIREN* also attains higher maj@k and avg@k on the smaller model Qwen2.5-Math-1.5B and the comparatively weaker model LLaMa3.1-8B. Through an analysis of pass@k and perplexity, which are commonly used to estimate the upper bound of reasoning and the uncertainty of generation (Holtzman et al., 2019; Yue et al., 2025; Cheng et al., 2025), we observe that *SIREN* yields more diverse responses and explores a wider scope, pushing the boundary of reasoning. In addition, the joint dynamics of entropy and validation pass@k during training indicate that *SIREN* performs effective, progressive exploration: it maintains high entropy in early stages and gradually converges later, ensuring sustained exploration while preventing degradation in validation pass@k.

In summary, the key contributions of this work are as follows.

- We analyze the limitations of naive entropy regularization in RLVR, showing that *the vast action space and long trajectories* flatten probability distributions across most positions. This highlights the need to control the effective scope of regularization.(Sec. 2)
- We propose *SIREN*, a novel method that selects exploration scopes at both the action and the trajectory levels for effective entropy regularization, while transforming the naive regularization into a self-anchored form to stabilize training.(Sec. 3)
- We achieve competitive results on mathematical benchmarks, showing consistent improvements over other entropy-related baselines on maj@k and avg@k.(Sec. 4)

## 2 PRELIMINARY ANALYSIS

As effective exploration remains a central challenge in traditional RL, entropy regularization is commonly employed to encourage policy diversity, thus facilitating broader state exploration (Haarnoja et al., 2018a;c; Liu et al., 2020). Mathematically, given a query q, the entropy of the current policy $\pi_\theta$ for each token $v_j$ is defined in Eq. 1, where $\mathcal{V}$ denotes the vocabulary.

$$\mathcal{H}_j = -\sum_{v \in \mathcal{V}} \pi_\theta(v \mid q, v_{<j}) \log \pi_\theta(v \mid q, v_{<j}).$$

(1)

The training objective is given by Eq 2, where $J_{PO}$ is the policy optimization objective and $\beta$ is the entropy coefficient controlling the strength of regularization. $o$ denotes a trajectory.

$$J = J_{PO} + \beta \frac{1}{|o|} \sum_{j=1}^{|o|} \mathcal{H}_j,$$

(2)

However, such naive entropy regularization exhibits notable limitations in the context of RLVR for LRMs. Cui et al. (2025) provides empirical evidence that it is highly sensitive to the entropy coefficient and prone to entropy collapse. The intuitive rationale behind this ineffectiveness lies in the large action spaces and long trajectories in LRMs, two characteristics that are markedly distinct from those of traditional RL tasks. In tasks like Hopper, Walker, or HalfCheetah, agents act in a 3–6 dimensional continuous space for a few hundred to about a thousand steps (Todorov et al., 2012). By contrast, LRMs select from hundreds of thousands of tokens over several thousand steps.

To better understand the phenomenon and validate our hypothesis, we conduct a preliminary experiment. Specifically, we train Qwen2.5-Math-7B with an entropy coefficient of 0.005, which is slightly higher than a suitable value for entropy regularization (as a reference, training with 0.001

**Questions**: Let $p$ be the least prime number for which there exists a positive integer $n$ such that $n^4 + 1$ is divisible by $p^2$. Find the least positive integer $m$ such that $m^4 + 1$ is divisible by $p^2$.

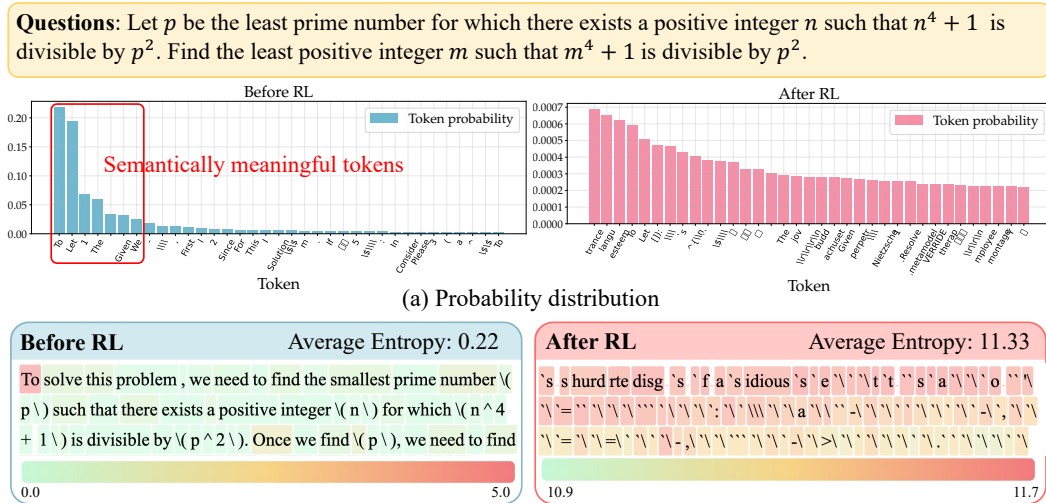

(a) Probability distribution

(b) Generated tokens with corresponding entropy visualization

Figure 1: Comparison of the first token's probability distributions and generated tokens before and after RL with naive entropy regularization (entropy coefficient set to 0.005).

does not lead to entropy explosion). All other datasets and hyperparameter configurations are kept consistent with the main experiment (Sec . 4.1). During training, we observe an entropy explosion at step 40 and use this checkpoint for comparison. We use a representative question from AIME24 (Li et al., 2024) and examine the probability distribution together with the generated tokens, qualitatively illustrating how the model behaves in the action space and over the trajectory, as shown in Figure 1. More cases can be found in Appendix G.

## 2.1 ANALYSIS OF TOKEN PROBABILITY DISTRIBUTIONS OVER ACTION SPACE

We first examine how RL with naive entropy regularization affects the action probabilities, i.e., the token probability distribution in LRMs. Figure 1(a) shows the distribution for the first token generated by the model before and after RL. Before RL, semantically meaningful tokens are concentrated at the top of the distribution, accounting for the vast majority of probability mass (red box in the left part of Figure 1(a)), while most other tokens are meaningless, with probabilities close to zero. After RL, however, the distribution becomes nearly uniform, with many semantically meaningless tokens rising to the top of the ranking. This phenomenon verifies our hypothesis: in LRMs, the extremely large action space and initially concentrated probabilities cause flattening of the distribution to yield disproportionately large entropy gains, reaching the maximum value of Eq. 1 and biasing the model toward increasing entropy.

**Policy Nucleus.** Motivated by this analysis, we introduce the concept of a policy nucleus to constrain exploration. The policy nucleus is defined as a subset of the vocabulary containing the most semantically important tokens, whose generation is reasonable and unlikely to cause incoherent responses. Since these tokens consistently occupy the top ranks in the original model's probability distribution, we adopt the terminology of Top-p sampling (Holtzman et al., 2019) and refer to this subset as the nucleus. Our key insight is that **exploration should be concentrated within the policy nucleus rather than uniformly across the entire vocabulary**, offering an effective solution to the challenges posed by the vast action space in LRMs.

## 2.2 ANALYSIS OF TOKEN ALONG LONG TRAJECTORIES

Next, we analyze how strong entropy regularization affects states along a long trajectory, i.e., the generated tokens in LRMs. As shown in Figure 1(b), before RL, the model produces coherent and meaningful solution steps with relatively low average token-level entropy. After RL, however, the outputs degenerate into largely meaningless token sequences, accompanied by a substantial increase in average entropy. To further illustrate this effect, we use heatmaps to visualize token-level policy entropy. In the model after RL, nearly every token exhibits uniformly high entropy, indicating that

the model explores extensively across almost all positions in the sequence. This may be because the autoregressive process turns early entropy spikes into a "chain reaction", i.e., uncertainty begets uncertainty, so as sequence length increases, the chance of such cascades grows, making explosions more likely.

In contrast, the original model shows variation in the token entropy across positions. Consistent with previous studies (Cui et al., 2025; Cheng et al., 2025; Wang et al., 2025), we observe that a small fraction of high-entropy tokens serve logical roles in guiding the reasoning process, such as *To, Once, Then*, while most tokens, which primarily fill sentences, exhibit much lower entropy. This observation suggests that **token-level entropy can be used to identify critical tokens, allowing regularization to be applied selectively**. Such targeted intervention may help prevent cascade spikes in a long trajectory of LRMs.

We provide theoretical grounding for the above phenomenon. From the perspective of gradient behavior, semantically meaningless tail tokens dominate the update direction, leading to either excessive exploration (i.e., gradient explosion) or ineffective exploration. The complete theoretical derivations can be seen in Appendix B.

## 3 SELECTIVE ENTROPY REGULARIZATION

Building on our preliminary analysis (Sec. 2), we introduce **SIREN** (SelectIve entRopy rEgularizatioN), a method that selectively applies entropy regularization to prevent the entropy explosion caused by large action spaces and long trajectories. **SIREN** employs a two-step entropy masking mechanism: (1) a *top-p mask* that restricts exploration to the policy nucleus, and (2) a *peak-entropy* mask that confines exploration to critical tokens (Sec.3.2). In addition, **SIREN** replaces the naive regularization with a *self-anchored regularization*, calibrating the regularization strength and improving training stability (Sec. 3.3). The overall framework is illustrated in Figure 2.

### 3.1 PRELIMINARY

Verifiable rewards remove the traditional reward model used in reinforcement learning and instead assign binary $0/1$ rewards by directly comparing the model's extracted answer with a predefined ground truth. Such rewards are widely adopted in tasks like mathematical reasoning and code generation, where correctness can be explicitly verified.

**GRPO.** Building on PPO (Schulman et al., 2017), Group Relative Policy Optimization (GRPO, Shao et al., 2024) models the entire response as a single action and computes the KL divergence over the full sequence rather than token by token. For each query $q$, GRPO samples a group of outputs $\{o_1, o_2, \ldots, o_G\}$ from the old policy $\pi_{\theta_{\text{old}}}$. Given the binary reward of output $o_i$ as $R(o_i)$, the group-normalized advantage $\hat{A}_i$ is defined as $\hat{A}_i = \frac{R(o_i) - \text{mean}\left(\{R(o_j)\}_{j=1}^G\right)}{\text{std}\left(\{R(o_j)\}_{j=1}^G\right)}$.

Then, inheriting from PPO, the training objective is:

$$J_{\text{GRPO}}(\theta) = \mathbb{E}_{q \sim \mathcal{P}(Q), \{o_i\}_{i=1}^G \sim \pi_{\theta_{\text{old}}}(O|q)}$$
$$\left[\frac{1}{G}\sum_{i=1}^G \frac{1}{|o_i|}\sum_{t=1}^{|o_i|} \min\left(\text{CLIP}(r_{i,t}(\theta), \epsilon)\hat{A}_{i,t}, r_{i,t}(\theta)\hat{A}_{i,t}\right) - \beta D_{\text{KL}}(\pi_\theta\|\pi_{\text{ref}})\right], \quad (3)$$

where $r_{i,t}(\theta) = \pi_\theta(o_{i,t}|q, o_{i,<t})/\pi_{\theta_{\text{old}}}(o_{i,t}|q, o_{i,<t})$ is an importance sampling term and $D_{KL}$ is the KL divergence.

**Dr.GRPO.** Dr.GRPO (Liu et al., 2025) modifies the original GRPO objective by removing the standard deviation term in the denominator of the advantage. The modified advantage is defined as $\hat{A}_i^{\text{Dr}} = R(o_i) - \text{mean}\left(\{R(o_j)\}_{j=1}^G\right)$, and correspondingly redefining the group-normalized advantage as:

$$J_{\text{Dr.GRPO}}(\theta) = \mathbb{E}_{q \sim \mathcal{P}(Q), \{o_i\}_{i=1}^G \sim \pi_{\theta_{\text{old}}}(O|q)}$$
$$\left[\frac{1}{G}\sum_{i=1}^G \sum_{t=1}^{|o_i|} \min\left(\text{CLIP}(r_{i,t}(\theta), \epsilon)\hat{A}_{i,t}^{Dr}, r_{i,t}(\theta)\hat{A}_{i,t}^{Dr}\right) - \beta D_{\text{KL}}(\pi_\theta\|\pi_{\text{ref}})\right], \quad (4)$$

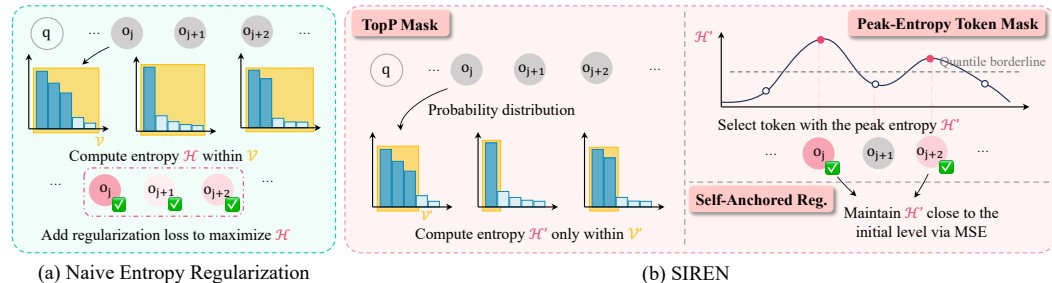

(a) Naive Entropy Regularization        (b) SIREN

Figure 2: The pipeline of **SIREN** compared to naive entropy regularization. (a) Naive entropy regularization computes entropy over the entire vocabulary and maximizes it uniformly across all tokens, which often leads to excessive exploration in LRMs. (b) **SIREN** introduces selective entropy regularization, including: (i) a *top-p mask* that restricts entropy computation to the policy nucleus, (ii) a *peak-entropy* token mask that identifies the most informative tokens, and (iii) a *self-anchored form of regularization* that maintains the entropy magnitude close to its initial level. Together, these components enable more targeted exploration and improved training stability.

Our method builds on Dr.GRPO, as experiments demonstrate that Dr.GRPO achieves higher accuracy while producing shorter responses compared to GRPO (Liu et al., 2025). We further enhance it with our proposed regularization techniques.

## 3.2 SELECTIVE TWO-STEP ENTROPY MASKING

We first apply top-p masking to recompute entropy within the policy nucleus for each token, and then apply peak-entropy token masking to identify critical tokens. For regularization, the entropy and gradients are computed only on these selected candidates and tokens.

**Top-p Mask within a Token.** Since LRMs have an extremely large action space, exploration over the entire space often assigns high probability to meaningless tokens, leading to incoherent outputs. To address this, we introduce a *top-p mask* to constrain exploration within the policy nucleus, i.e., the subset of meaningful tokens that are worth exploring. A visualization is shown in the yellow region on the left side of Figure 2(b). Formally, the policy nucleus is defined as:

$$\mathcal{V}_j^{(p)} = \arg \min_{\mathcal{S} \subseteq \mathcal{V}} |\mathcal{S}| \ s.t. \sum_{v \in \mathcal{S}} P(v \mid v_{<j}) \geq p, \tag{5}$$

where $\mathcal{V}$ denotes the original vocabulary, and $P(v \mid v_{<j})$ is the probability of token $v_j$.

The top-p mask and the corresponding recomputed entropy are then defined as:

$$M_j^{\text{top-p}}(v) = \begin{cases} 1, & v \in \mathcal{V}_j^{(p)} \\ 0, & \text{otherwise}, \end{cases} \quad P'(v \mid v_{<j}) = \frac{M_j^{\text{top-p}}(v) \, P(v \mid v_{<j})}{\sum_{u \in \mathcal{V}} M_j^{\text{top-p}}(u) \, P(u \mid v_{<j})}, \tag{6}$$

$$\mathcal{H}_j' = -\sum_{v \in V} P'(v \mid v_{<j}) \log P'(v \mid v_{<j}). \tag{7}$$

**Peak-Entropy Mask for a Trajectory.** Similarly, as analyzed in Sec.2, for long trajectories in LRMs, uncertainty accumulates exponentially, eventually causing entropy to explode at every token position, as exemplified in the right part of Figure1(b). Only a small subset of tokens plays a disproportionately important role in exploration, typically corresponding to sentence heads or logical connectors that determine the reasoning direction, while most other tokens serve as semantic fillers. Consistent with prior observations (Cheng et al., 2025; Wang et al., 2025; Cui et al., 2025), we find that such critical tokens tend to exhibit relatively high entropy. To focus on regularization on these tokens, we introduce a *peak-entropy mask*, which selects tokens whose entropy lies in the top $\tau$-quantile within a trajectory. Concretely, for the $i$-th trajectory $o_i$ in a batch, we define the $\tau$-quantile of the token-level entropy, computed from the recomputed entropy $\mathcal{H}'$ in Eq. 7, as:

$$q_\tau^{(i)} = \text{Quantile}_\tau \left\{ \mathcal{H}'(v_1^{(i)}), \mathcal{H}'(v_2^{(i)}), \ldots, \mathcal{H}'(v_{|o_i|}^{(i)}) \right\}, \tag{8}$$

and construct the peak-entropy mask $\mathbf{m}_j^{\text{peak}}$ by retaining tokens above the $\tau$-quantile of entropy:

$$\mathbf{m}_j^{\text{peak}} = \begin{cases} 1, & \text{if } \mathcal{H}'(v_j) \geq q_\tau^{(i)}, \\ 0, & \text{otherwise.} \end{cases} \quad (9)$$

Finally, the entropy $\overline{\mathcal{H}}$ is aggregated over all trajectories within a batch as shown in Eq. 10, where $G$ is the number of trajectories in the batch.

$$\overline{\mathcal{H}} = \frac{1}{\sum_{i=1}^{G} \sum_{j=1}^{|o_i|} \mathbf{m}_j^{peak}} \sum_{i=1}^{G} \sum_{j=1}^{|o_i|} \mathbf{m}_j^{\text{peak}} \cdot \mathcal{H}'\big(v_j^{(i)}\big). \quad (10)$$

### 3.3 Self-Anchored Regularization

The naive objective of entropy regularization, which generally aims to maximize entropy (Haarnoja et al., 2018a; Liu et al., 2020), can easily fall into two extremes in the context of LRMs: either excessively high, leading to global explosion and incoherent outputs, or too low, failing to mitigate premature convergence of the policy. To address this issue and keep the policy's entropy at an appropriate level, we modify the objective into a self-anchored regularization. It is defined as the mean squared error (MSE) between the aggregated entropy $\overline{\mathcal{H}}$ (computed as in Eq. 10) and an entropy anchor $\mathcal{H}_a$:

$$L_{sa} = \big(\overline{\mathcal{H}} - \mathcal{H}_a\big)^2. \quad (11)$$

Accordingly, the new training objective is formulated as $J' = J_{PO} - \beta L_{sa}$. Instead of introducing an external hyperparameter for the entropy anchor $\mathcal{H}_a$, we initialize it using the aggregated entropy $\overline{\mathcal{H}}$ at the initial step (Eq. 10), which is why we term it *self-anchored*. This initialization is model- and dataset-specific, reflecting the inherent uncertainty of the pre-trained policy before accuracy-driven RL fine-tuning. At this stage, the model typically produces more diverse outputs with higher entropy, providing a natural and exploration-friendly starting point.

## 4 Experiment

### 4.1 Setup

**Models.** The main experiments and analysis are conducted on Qwen2.5-Math-7B (Yang et al., 2024). We also extend our method to the smaller model Qwen2.5-Math-1.5B (Yang et al., 2024). and the weaker model Llama3.1-8B (Dubey et al., 2024), with results analyzed in Sec. 4.2.

**Training Configurations.** For the Qwen series models, following the setup in LUFFY (Yan et al., 2025), we train on the OpenR1-Math-46k-8192 dataset[1], a subset of OpenR1-Math-220k (Face). Consistent with DAPO (Yu et al., 2025), we remove the KL loss term and increase the clip ratio to 0.28. The rollout batch size is set to 128, and the update batch size is 8. We perform 8 rollouts per prompt and set the temperature to 1.0 for rollout generation. For Llama3.1-8B, given the relative weakness of the model, we use a simpler dataset constructed from the OpenR1-Math-46k-8192, GSM8K (Cobbe et al., 2021), and MATH500 (Hendrycks et al., 2021) training sets. When training Llama3.1-8b, we use 16 rollouts per prompt, with a rollout batch size of 512 and an update batch size of 32. Additional details on dataset construction and training can be found in Appendix C.

**Evaluation.** For evaluation, we focus on five widely used mathematical reasoning benchmarks, namely AIME24, AIME25, AMC (Li et al., 2024), OlympiadBench (He et al., 2024), and MATH500 (Hendrycks et al., 2021). We report maj@k and avg@k. Maj@k (majority voting, Wang et al., 2022) selects the most frequent answer among k samples and verifies its correctness. This metric does not require an external verifier and can simultaneously capture both the model's exploration ability and its inherent confidence. Avg@k, which averages scores over k responses, is a commonly used metric that captures the model's overall performance (Yu et al., 2025; Yan et al., 2025). For AIME24, AIME25, and AMC, we set n to 32 due to the relatively small test sets; for the remaining benchmarks, we set n to 8. The temperature is fixed at 0.6 for all evaluations. For each model, we select the checkpoint with the highest validation accuracy for evaluation, where validation is performed every 10 steps; detailed settings are provided in the Appendix C.

---

[1]https://huggingface.co/datasets/Elliott/Openr1-Math-46k-8192

Table 1: Overall performance of Qwen2.5-Math-7B compared with Dr.GRPO, naive entropy regularization, and other RLVR methods that also incorporate entropy. All models are evaluated under a unified setting. Bold indicates the best results.

| | AIME24 | | AIME25 | | AMC22 | | MATH500 | | Olympiad Bench | | Average | |
| --- | --- | --- | --- | --- | --- | --- | --- | --- | --- | --- | --- | --- |
| | *Maj@32* | *Avg@32* | *Maj@32* | *Avg@32* | *Maj@32* | *Avg@32* | *Maj@8* | *Avg@8* | *Maj@8* | *Avg@8* | *Maj.* | *Avg.* |
| Qwen2.5-Math-7B | 30.0 | 11.0 | 13.3 | 3.8 | 60.2 | 29.8 | 70.8 | 46.4 | 25.3 | 15.4 | 39.9 | 21.3 |
| Dr.GRPO | 30.0 | 21.5 | 16.7 | 15.4 | 67.5 | 61.1 | 84.2 | 81.2 | 45.3 | 42.0 | 48.7 | 44.2 |
| Naive Entropy Reg. | 33.3 | 28.0 | 20.0 | 14.1 | 63.9 | 58.9 | 84.2 | 80.4 | 44.4 | 40.9 | 49.2 | 44.5 |
| Dual Gradient Reg. | 36.7 | **30.2** | 20.0 | 16.7 | 68.7 | **63.7** | 82.8 | 79.6 | 43.7 | 40.7 | 50.4 | **46.2** |
| Clip-Cov | 33.3 | 22.5 | 20.0 | **18.1** | 67.5 | 59.3 | 78.4 | 76.6 | 41.6 | 38.3 | 48.2 | 43.0 |
| KL-Cov | 36.7 | 26.7 | 16.7 | 15.1 | 66.3 | 60.2 | 84.0 | 79.6 | 44.0 | 40.9 | 49.5 | 44.5 |
| Entropy Adv. | 30.0 | 27.4 | 13.3 | 12.9 | 65.1 | 58.7 | 80.8 | 78.4 | 43.0 | 40.0 | 46.4 | 43.5 |
| On Forking Tokens | 36.7 | 24.0 | 20.0 | 13.9 | 62.7 | 57.2 | 84.6 | 81.1 | 45.0 | 40.9 | 49.8 | 43.3 |
| *SIREN* (ours) | **43.3** | 28.4 | **26.7** | 17.9 | **71.1** | 60.1 | **85.0** | 81.9 | **46.8** | 42.5 | **54.6** | 46.1 |

**Baseline methods.** We implement several prior approaches that incorporate entropy during reinforcement learning as baselines for comparison, including: (1) *Dr.GRPO (Liu et al., 2025)*; (2) *Naive Entropy Regularization* with an entropy coefficient of 0.001; (3) *Dual Gradient Regularization (Haarnoja et al., 2018c)*, an online variant of SAC with automatically adjusted temperature that updates $\alpha$ by gradient ascent on $J(\alpha) = \mathbb{E}[-\alpha H_t - \alpha H_{\text{target}}]$; (3) *Clip-Cov / KL-Cov (Cui et al., 2025)*: mitigating entropy collapse by either clipping tokens that exhibit high covariance between action probability and advantage, or by applying KL penalties to those tokens; (4) *Entropy Adv. (Cheng et al., 2025):* augmenting the advantage function with an entropy-based term; (5) *RL on forking tokens (Wang et al., 2025)*: applying policy gradient updates only to high-entropy tokens. More detailed motivation and implementation specifics of this baseline are provided in Appendix C.2.

## 4.2 MAIN RESULTS

**SOTA performance on RLVR with Qwen2.5-Math-7B.** Table 1 presents a comparative evaluation of the original Qwen2.5-Math-7B model, several baseline methods, and our proposed approach. *SIREN* yields substantial gains over the base model and achieves competitive or superior performance compared to strong RL baselines across benchmarks. Specifically, it achieves an average maj@k of **54.6**, outperforming the strongest baseline by **+4.8** points. In AIME24 and AIME25, the most challenging benchmarks, our method achieves a maj@32 score of **43.3** and **26.7**, respectively, achieving consistent improvements of **+6.6** points over the best-performing baseline on both datasets. For the overall average avg@n, our method reaches **46.1**, surpassing previous approaches and setting a new record.

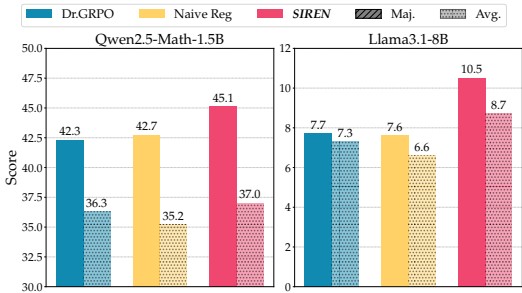

Figure 3: Average maj@k and avg@k across five mathematical benchmarks for *SIREN* on different backbones, including Qwen2.5-Math-1.5B and LLaMa3.1-8B (see Appendix E.1 for detailed results).

**Extending to Other Models.** To further verify the generalizability of our method, we extend *SIREN* to smaller and weaker models. The evaluation results are presented in Figure 3. On the smaller model Qwen2.5-math-1.5B and the weaker model LLaMA3.1-8B, *SIREN* consistently achieves the best maj@k and avg@k across five mathematical benchmarks, outperforming both Dr.GRPO and naive entropy regularization. *SIREN* improves maj@k by +2.4 on Qwen2.5-math-1.5B and +2.8 on LLaMA3.1-8B.

## 5 ANALYSIS

We conduct a detailed analysis to investigate the relationship between *SIREN* and the exploration capability of LRMs, from both theoretical and experimental perspectives. First, we provide a theoretical analysis of *SIREN*'s masking mechanism, explaining how it mitigates both gradient explosion and collapse. From the experimental perspective, we evaluate the pass@k performance of *SIREN* (Figure 4), which illustrates its effectiveness in promoting exploration and expanding the reasoning boundary. We then compute the perplexity of responses generated by *SIREN* (Figure 5), showing that it preserves greater diversity than naive methods, thereby facilitating exploration. Next,

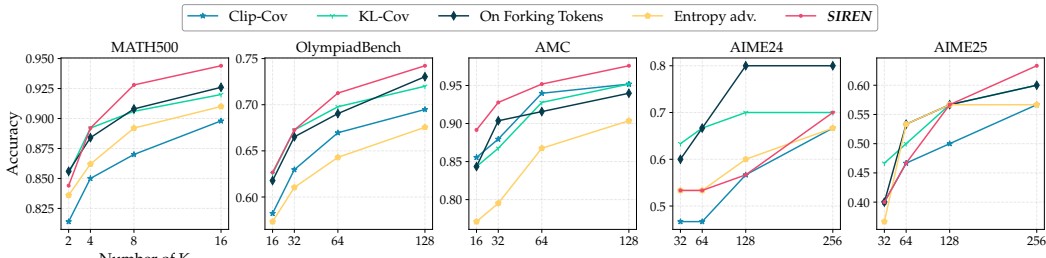

Figure 4: Pass@k performance of *SIREN* in comparison with various RLVR baselines, scaling with k.

we examine the training dynamics, including entropy variations and validation pass@16 (Figure 6), which further confirm *SIREN*'s ability to control exploration. Finally, we conduct an ablation study to assess the contribution of each module, as well as a sensitivity analysis to evaluate the impact of key hyperparameters on model performance.

## 5.1 THEORETICAL GROUNDING OF *SIREN*

Naive entropy regularization either suffers from **gradient explosion**, due to large contributions from low-probability tail tokens, or produces **ineffective exploration**, because most gradient energy on meaningful head tokens becomes negligible for large vocabularies. *SIREN* addresses both issues via two complementary masks: a token-level top-$p$ mask removes tail tokens, bounding per-step entropy gradients and concentrating exploration on semantically meaningful actions, while a trajectory-level peak-entropy mask applies entropy only at high-uncertainty positions, preventing the entropy objective from dominating the reward.

Together, these masks also substantially reduce the **variance of entropy-gradient estimates**. By eliminating noisy contributions from tail tokens and focusing the entropy objective on a limited set of high-entropy steps, SIREN produces localized, low-variance gradients whose second moments scale with the much smaller nucleus size $K \ll V$ rather than the full vocabulary V. As a result, the policy receives more stable and meaningful gradient signals, enabling both **stable training** and **effective exploration**, thereby resolving the fundamental trade-off faced by naive entropy regularization. The detailed derivation and formal analysis are provided in Appendix B.

## 5.2 *SIREN* PROMOTES EXPLORATION

**Pass@k Analysis.** We adopt pass@k to examine the exploration boundary of models trained with *SIREN*. Following prior work (Yue et al., 2025; Cheng et al., 2025), pass@k is widely recognized as an indicator of the upper bound of reasoning ability. Specifically, we set k to 256 for the difficult yet small-scale benchmarks (AIME, AIME 2025), k to 128 for the medium-level and relatively larger benchmarks (AMC, OlympiadBench), and k to 16 for the simpler benchmark (MATH500). The results are presented in Figure 4. *SIREN* achieves strong performance even at small values of k, with its advantage increasing as k grows, particularly on relatively easy benchmarks such as MATH500, AMC, and Olympiadbench. On the hardest benchmark, AIME25, *SIREN* achieves the best pass@k at the maximum k. On AIME24, however, it performs slightly worse than some baselines in pass@k while achieving higher maj@k and avg@k, as shown in Table 1. These results demonstrate that *SIREN* not only explores a wider range of possible responses, but also effectively balances exploration with exploitation, producing higher-quality and more consistent answers across multiple trials.

**Perplexity Analysis.** Besides pass@k, the exploratory effects promoted by *SIREN* can also be verified using perplexity (PPL). Following Yue et al. (2025), PPL measures a model's ability to predict a response, with lower PPL indicating higher probability to generate response $Y$. Previous work (Yue et al., 2025) shows that RL reduces PPL, revealing a collapse in a smaller subset of responses and thus harming exploration. We quantitatively evaluated the original model's PPL of responses generated by the original model, Dr.GRPO, naive entropy regularization, and *SIREN*. For responses, we reused the outputs obtained during the evaluation phase. As shown in Fig. 5, *SIREN*

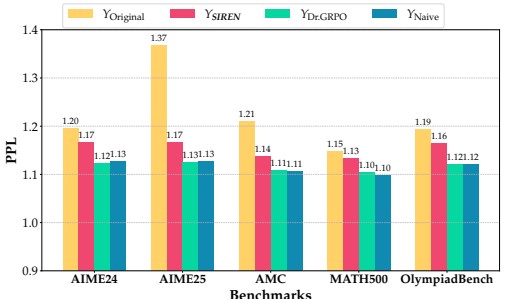 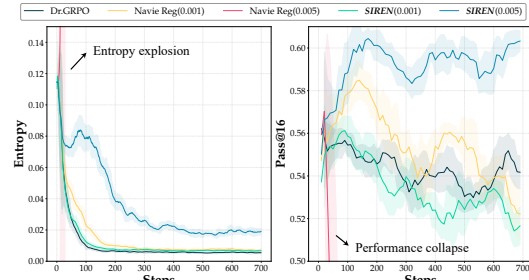

Figure 5: Average perplexity of generated responses across all benchmarks, computed using the base model Qwen2.5-Math-7B.

Figure 6: Training dynamics of entropy (left) and pass@16 performance (right) under different regularization strategies.

consistently achieves higher PPL compared to baseline RL methods, suggesting that it effectively mitigates the convergence of responses and encourages exploration over a broader range of possible outputs.

### 5.3 *SIREN* MAKES EXPLORATION MORE EFFECTIVE

We further study the training dynamics to understand why *SIREN* can promote exploration. We focus on two metrics closely related to exploration, namely entropy and validation pass@16, with the results shown in Figure 6. Dr.GRPO, Naive Reg (0.001), and *SIREN* (0.001) all exert little to no control over entropy, causing the policy to converge prematurely during the early stages of training. This lack of entropy regulation limits exploration of new states, leading to a decrease in validation pass@k. In contrast, Naive Reg (0.005) uses an excessively large entropy coefficient, resulting in rapid entropy explosion and poor performance. *SIREN* (0.005), however, maintains a substantial level of entropy during the early training stage, with a slight increase between steps 50–80, which encourages the generation of more diverse responses. As training progresses, the entropy gradually converges and eventually stabilizes at a higher level than the other baselines. By keeping entropy at an appropriate level throughout training, *SIREN* ensures continuous exploration, which is effectively converted into more correct responses and leads to improved validation pass@k. Comparison of *SIREN* with other baselines is provided in the Appendix F.2.

### 5.4 ABLATION STUDY

Our method consists of three components: *the peak-entropy mask, the top-p mask, and the self-anchored regularization* We perform ablations by removing each component in turn. The average results are shown in Table 2. In these benchmarks, removing any single component from our method leads to noticeable drops in both maj@k and avg@k. In particular, removing the *self-anchored regularization* causes the most severe degradation, with maj@k and avg@k decreasing by **+10.3** and **+15.5** points,

Table 2: Abaltion results of *SIREN* on Qwen2.5-Math-7B.

|  | *Maj.* | *Avg.* |
|---|---|---|
| ***SIREN*** | **54.6** | **46.1** |
| w/o TopP Mask | 49.2 | 43.1 |
| w/o Peak-Entropy Mask | 48.9 | 44.0 |
| w/o Self-Anchored Reg. | 44.3 | 30.6 |
| w/ only Self-Anchored Reg. | 52.3 | 43.6 |

respectively. This is because the self-anchored regularization balances performance and entropy: it sustains high entropy for exploration while maintaining stability. Without it, even when we set the coefficient to a very small value 0.0001, the entropy still rises to an excessively high level, leading to a severe drop in performance. For the *peak-entropy mask* and the *Top-p mask*, removing either leads to a drop in performance, indicating that each mask alone is insufficient to fully regulate the model's exploration, while their combined usage yields the best results. Given the severe performance drop, we conduct an ablation study using *only self-anchored regularization*. This demonstrates that the stabilization technique can improve the performance of naive entropy regularization. However, its effectiveness remains inferior to that of *SIREN*. In particular, on the AIME24 dataset, using only self-anchored regularization yields 30.0 maj@k and 22.3 avg@k, compared to *SIREN*'s 43.3 and 28.4, indicating a significant drop. Hyperparameters and full experimental results are provided in Appendix E.2.

Table 3: Sensitivity analysis of *SIREN* with respect to the top-p cutoff p and peak-entropy quantile $\tau$.

|  | p = 0.8, $\tau$=0.8 | p = 0.7 | p = 0.9 | $\tau$ = 0.7 | $\tau$ = 0.75 | $\tau$ = 0.85 | $\tau$ = 0.9 |
|---|---|---|---|---|---|---|---|
| Avg.Maj | **54.6** | 48.5 | 52.0 | 50.9 | 54.1 | 52.6 | 50.3 |
| Avg.Avg | **46.1** | 42.2 | 44.8 | 43.5 | 46.0 | 43.7 | 40.7 |

## 5.5 SENSITIVITY ANALYSIS

In this section, we investigate the sensitivity of *SIREN* to its two key hyperparameters: the top-p cutoff and the peak-entropy quantile $\tau$. Starting from the default setting of 0.8 for both parameters, we vary each hyperparameter while keeping the other fixed in order to assess how changes in these thresholds affect model performance. As shown in the table 3,increasing the top-p cutoff value leads to only a slight degradation in model performance. However, when the value is reduced to 0.7, performance drops significantly. This is likely because a cutoff of 0.7 is too small in terms of probability; many tokens end up with zero entropy after applying the top-p mask, causing them to be filtered out. With too few tokens left for entropy regularization to take effect, the model's performance suffers. Regarding $\tau$, adjusting it by ±0.05 results in stable performance, and even a reduction of 0.1 maintains acceptable avg@k. In contrast, increasing $\tau$ by 0.1 leads to a more substantial performance decline. This may be attributed to excessively large quantiles filtering out effective logical tokens. Overall, adjusting p and $\tau$ within a reasonable range does not cause severe performance degradation, indicating that SIREN is robust and does not rely heavily on precise hyperparameter tuning. Detailed experimental results are provided in Appendix E.3.

## 6 RELATED WORK

**Reinforcement Learning for LLMs.** Reinforcement learning has been a major technique for post-training large language models (LLMs) (Ouyang et al., 2022; Touvron et al., 2023; Rafailov et al., 2023). Recently, RLVR, which replaces the reward model with a verification function that compares the model's boxed answer against the ground truth, has demonstrated substantial potential for improving reasoning capabilities (Shao et al., 2024; DeepSeek-AI, 2025; Jaech et al., 2024; Team et al., 2025). This has attracted increasing attention from the research community, leading to a variety of subsequent improvements (Zeng et al., 2025; Hu et al., 2025; Yeo et al., 2025; Liu et al., 2025; Yu et al., 2025). Yet, despite its promise, RLVR still faces significant challenges such as entropy collapse and premature convergence, which severely restrict exploration and limit the performance.

**Exploration in reinforcement learning.** Effective exploration remains an important topic in reinforcement learning. One line of research incorporates intrinsic motivation or reward bonuses to construct auxiliary signals that incentivize agents to explore new states (Bellemare et al., 2016; Ostrovski et al., 2017; Pathak et al., 2017; Burda et al., 2018; Fortunato et al., 2019). Another line of work improves exploration by maximizing entropy, thereby introducing uncertainty into the policy to encourage more diverse behaviors (Ziebart, 2010; Haarnoja et al., 2017; 2018a;c). As an important signal for policy optimization, entropy has also been utilized in RLVR. Cui et al. (2025) investigates the empirical correlation between entropy and performance, and proposes clipping or KL-based mechanisms to indirectly control entropy. Meanwhile, Cheng et al. (2025); Zheng et al. (2025); Wang et al. (2025) treat entropy as a heuristic signal, using it for advantage shaping, enhancing the rollout phase, or loss masking. Unlike the aforementioned methods, which exploit entropy only indirectly or as an auxiliary signal, our method analyzes the failure of naive entropy regularization and directly extends it from traditional RL to RLVR for LRMs. This enables explicit and effective control of entropy, thereby directly promoting policy exploration.

## 7 CONCLUSION

This work revealed that entropy collapse in LRMs arises from uncontrolled exploration, amplified by vast action spaces and long trajectories. To address this, we introduced *SIREN*, which increases naive regularization with a two-step entropy mask and self-anchored regularization. Experiments and analysis demonstrated that *SIREN* not only stabilizes training but also drives richer reasoning, surpassing previous entropy-based methods. Our results underlined a central insight: LRMs require targeted and effective exploration, providing a foundation for future advances in enhancing LRM exploration.

## ETHICS STATEMENT

We are not aware of any major ethical concerns arising from our work. Our study is conducted entirely within the mathematics domain, using only publicly available models and datasets for training and evaluation. No human subjects were involved, and our research does not introduce sensitive or potentially harmful insights.

## REPRODUCIBILITY STATEMENT

We have made extensive efforts to ensure the reproducibility of our work. Our RL algorithm is implemented based on verl(Sheng et al., 2025), with modifications to the entropy calculation, entropy aggregation, and objective computation. The complete implementation is provided in an anonymous repository: `https://anonymous.4open.science/r/siren_anonymous-6888`. All models and datasets used in this work are open source. Detailed information about the training setup, including datasets, prompts, hyperparameters, and GPU configurations, is provided in Sec.4.1 and Appendix C and D. We believe these measures will enable other researchers to reproduce our work and further advance the field.

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

## A    LLM USAGE

During the preparation of this work, we used GPT-5 solely to improve the language. The LLMs served only as assistant tools and did not contribute to any intellectual or scientific aspect of the work. After using this tool, we carefully reviewed and edited all content and take full responsibility for the publication.

## B    THEORETICAL ANALYSIS

In this section, we provide the theoretical grounding of *SIREN*'s masking mechanism. We first analyze why naive entropy regularization fails by examining its gradient structure and inherent instability, and how *SIREN* mitigates these issues. We then show that *SIREN* reduces entropy variance, leading to more stable and effective entropy optimization.

### B.1    GRADIENT-BASED EXPLANATION

We analyze why naive entropy regularization is either **unstable** (entropy explosion) or **ineffective** (negligible exploration), and how *SIREN*'s two masks (a token-level top-$p$ mask and a trajectory-level peak-entropy mask) simultaneously **bound entropy growth** and **concentrate exploration on meaningful actions and positions**.

#### B.1.1    WHY NAIVE ENTROPY REGULARIZATION FAILS IN RLVR FOR LRM

Let $\mathcal{V}$ be the vocabulary with size $|\mathcal{V}| = V$, trajectory length $T$, and policy logits $z_t$ at position $t$.

The naive entropy term at step $t$ is

$$H_t(\theta) = -\sum_{v \in \mathcal{V}} p_t(v) \log p_t(v), \tag{12}$$

with gradient (up to the softmax Jacobian)

$$\frac{\partial H_t}{\partial z_t(v)} = -\big(1 + \log p_t(v)\big)\frac{\partial p_t(v)}{\partial z_t(v)}. \tag{13}$$

**Head–Tail Decomposition**    For each step $t$, define:

- **Head (nucleus)** $\mathcal{V}_t^{\text{head}}$: smallest set of tokens with cumulative mass $\geq p$; size $K_t \leq K$.
- **Tail** $\mathcal{V}_t^{\text{tail}} = \mathcal{V} \setminus \mathcal{V}_t^{\text{head}}$.

**Assumption 1 (matches behavior of pretrained LLMs)** *There exist constants $c_{\text{head}}, c_{\text{tail}} > 0$ such that:*

$$\sum_{v \in \mathcal{V}_t^{\text{head}}} p_t(v) \geq c_{\text{head}}, \qquad p_t(v) \leq \frac{c_{\text{tail}}}{V - K_t} \quad \textit{for all } v \in \mathcal{V}_t^{\text{tail}}.$$

**Lemma 1 (Naive entropy gradient is dominated by tail coordinates)** *Under Assumption 1, there exist constants $A_{\text{head}}, A_{\text{tail}} > 0$ such that:*

$$\sum_{v \in \mathcal{V}_t^{\text{head}}} \left\|\frac{\partial H_t}{\partial z_t(v)}\right\|^2 \leq A_{\text{head}} K_t, \tag{14}$$

$$\sum_{v \in \mathcal{V}_t^{\text{tail}}} \left\|\frac{\partial H_t}{\partial z_t(v)}\right\|^2 \geq A_{\text{tail}}(V - K_t)\log^2 V. \tag{15}$$

Thus, naive entropy focuses its gradient on *tail tokens*, which are semantically meaningless.

**Explosion vs. Ineffectiveness**  Let reward be bounded by $|R(o)| \leq R_{\max}$. The naive objective is

$$J_{\text{naive}}(\theta) = \mathbb{E}[R(o)] + \beta \sum_{t=1}^{T} H_t(\theta). \tag{16}$$

- **Entropy explosion.** Entropy per step is at most $\log V$. Total entropy reward is at most $\beta T \log V$. If $\beta \log V \gg R_{\max}$, entropy dominates the reward and pushes the policy toward uniform, causing explosion, especially since $T$ is large.
- **Ineffective exploration.** To avoid explosion, $\beta$ must be small. But then the fraction of entropy-gradient energy on the head is at most $\frac{A_{\text{head}} K}{A_{\text{tail}}(V-K)\log^2 V}$, which becomes negligible for large $V$. Thus, *naive entropy barely changes head tokens* and yields almost no exploration.

### B.1.2  WHY *SIREN* ENABLES STABLE AND EFFECTIVE ENTROPY REGULARIZATION

***SIREN*'s Token-Level Top-$p$ Mask**  *SIREN* replaces $H_t$ with the **nucleus entropy**

$$H_t^{(p)} = - \sum_{v \in \mathcal{V}_t^{\text{head}}} p_t^{(p)}(v) \log p_t^{(p)}(v), \tag{17}$$

where $p_t^{(p)}$ renormalizes $p_t$ over $\mathcal{V}_t^{\text{head}}$.

Then:

$$\frac{\partial H_t^{(p)}}{\partial z_t(v)} = \begin{cases} \text{entropy gradient}, & v \in \mathcal{V}_t^{\text{head}}, \\ 0, & v \in \mathcal{V}_t^{\text{tail}}. \end{cases}$$

Thus:

1. **Tail gets zero entropy gradient.** This removes the instability caused by pushing thousands of meaningless tail tokens.
2. **Entropy per step is bounded by** $\log K$**.**
$$H_t^{(p)} \leq \log K.$$
3. **Exploration on head tokens becomes strong and meaningful.** All entropy gradient is concentrated on $\leq K$ tokens. Therefore, for the same $\beta$, exploration on useful actions increases dramatically.

***SIREN*'s Peak-Entropy Mask**  Let

$$\mathcal{T}_{\text{peak}} = \{t : H_t^{(p)} \text{ in top-}\rho \text{ fraction}\}, \qquad |\mathcal{T}_{\text{peak}}| = \rho T.$$

*SIREN* applies entropy only at these steps:

$$J_{\textbf{\textit{SIREN}}}^{\text{entropy}}(\theta) = \beta \sum_{t \in \mathcal{T}_{\text{peak}}} H_t^{(p)}(\theta). \tag{18}$$

1. **Entropy cannot dominate reward.**
$$J_{\textbf{\textit{SIREN}}}^{\text{entropy}} \leq \beta \rho T \log K.$$
2. **Stronger exploration at selected positions.** Naive entropy gives each step a scale $\sim \beta/T$; *SIREN* gives $\sim \beta/(\rho T)$ on selected steps. Local exploration signals are therefore amplified by $1/\rho$.
3. **Low-entropy positions contribute almost no useful exploration.**
   If entropy is small, say $H_t \leq H_{\text{low}}$, then most mass lies on a single token $v_t^\star$:
$$p_t(v_t^\star) \geq 1 - \delta(H_{\text{low}}), \qquad \delta(H_{\text{low}}) \to 0.$$
   Then the naive entropy gradient satisfies
$$\sum_{v \in \mathcal{V}} \left\| \beta \frac{\partial H_t}{\partial z_t(v)} \right\|^2 \leq C \beta^2 \delta(H_{\text{low}}),$$
   which is negligible for a safe small $\beta$.

Thus, naive entropy has almost no effect at low-entropy positions. *SIREN* simply removes them, losing nothing while reducing instability.

## B.2 Variance Reduction Advantages of *SIREN*

In this section we show that *SIREN* yields *lower-variance* entropy gradients than naive entropy regularization for the same coefficient $\beta$, in the large-vocabulary regime.

Let $g_{\text{naive}}(\theta; o)$ and $g_{\text{SIREN}}(\theta; o)$ denote the entropy-gradient contributions to the policy gradient for a single rollout $o$, under naive entropy and *SIREN* respectively. We focus on their variance over trajectories $o \sim \pi_\theta$:

$$\text{Var}[g] := \mathbb{E}\big[\|g - \mathbb{E}g\|^2\big] = \mathbb{E}\big[\|g\|^2\big] - \|\mathbb{E}g\|^2.$$

### B.2.1 Per-step squared norms

Recall from the previous lemmas that at each step $t$:

$$\|g_t^{\text{naive}}\|^2 = \beta^2 \sum_{v \in \mathcal{V}} \left\|\frac{\partial H_t}{\partial z_t(v)}\right\|^2 \geq \beta^2 A_{\text{tail}}(V - K_t)\log^2 V, \tag{19}$$

$$\|g_t^{\text{SIREN}}\|^2 = \beta^2 \sum_{v \in \mathcal{V}_t^{\text{head}}} \left\|\frac{\partial H_t^{(p)}}{\partial z_t(v)}\right\|^2 \leq \beta^2 B_{\text{head}}K_t, \tag{20}$$

for constants $A_{\text{tail}}, B_{\text{head}} > 0$ independent of $V$ and $T$. In particular, $g_t^{\text{SIREN}}(v) = 0$ for all $v \in \mathcal{V}_t^{\text{tail}}$.

We write the trajectory-level entropy gradients approximately as

$$g_{\text{naive}}(\theta; o) \approx \sum_{t=1}^{T} g_t^{\text{naive}}(\theta; o), \qquad g_{\text{SIREN}}(\theta; o) \approx \sum_{t \in \mathcal{T}_{\text{peak}}(o)} g_t^{\text{SIREN}}(\theta; o),$$

where $\mathcal{T}_{\text{peak}}(o)$ is the set of peak-entropy positions for trajectory $o$, with $|\mathcal{T}_{\text{peak}}(o)| = \rho T$.

We make a mild aggregation assumption that per-step entropy gradients are not perfectly canceling.

**Assumption 2 (Weak temporal correlation)** *There exist constants $c_1, c_2 > 0$ (independent of $V, T$) such that*

$$\mathbb{E}\big[\|g_{\text{naive}}\|^2\big] \geq c_1 \sum_{t=1}^{T} \mathbb{E}\big[\|g_t^{\text{naive}}\|^2\big], \tag{21}$$

$$\mathbb{E}\big[\|g_{\text{SIREN}}\|^2\big] \leq c_2 \sum_{t \in \mathcal{T}_{\text{peak}}} \mathbb{E}\big[\|g_t^{\text{SIREN}}\|^2\big]. \tag{22}$$

This simply requires that cross-terms in $\mathbb{E}\|\sum_t g_t\|^2$ do not systematically cancel the per-step squared norms.

Using equation 19–equation 20 and Assumption 2, and noting that $K_t \leq K$ and $V - K_t \geq V - K$, we obtain

$$\mathbb{E}\big[\|g_{\text{naive}}\|^2\big] \geq c_1 \sum_{t=1}^{T} \beta^2 A_{\text{tail}}(V - K_t)\log^2 V \geq c_1 \beta^2 A_{\text{tail}} T(V - K)\log^2 V, \tag{23}$$

$$\mathbb{E}\big[\|g_{\text{SIREN}}\|^2\big] \leq c_2 \sum_{t \in \mathcal{T}_{\text{peak}}} \beta^2 B_{\text{head}}K_t \leq c_2 \beta^2 B_{\text{head}}\rho T K. \tag{24}$$

For large vocabulary $V$ and modest nucleus size $K \ll V$, the ratio of second moments satisfies

$$\frac{\mathbb{E}\big[\|g_{\text{naive}}\|^2\big]}{\mathbb{E}\big[\|g_{\text{SIREN}}\|^2\big]} \geq \frac{c_1 A_{\text{tail}}}{c_2 B_{\text{head}}} \cdot \frac{V - K}{\rho K} \cdot \log^2 V, \tag{25}$$

which grows without bound as $V \to \infty$.

### B.2.2 FROM SECOND MOMENT TO VARIANCE

We now relate the second moments to the variances.

**Assumption 3 (Bounded mean gradient)** *There exists $M > 0$, independent of $V$ and $T$, such that*

$$\big\|\mathbb{E}[g_{\mathrm{naive}}]\big\| \leq M, \qquad \big\|\mathbb{E}[g_{\mathrm{SIREN}}]\big\| \leq M.$$

This simply states that the expected entropy gradient (the bias term) does not grow with vocabulary size; it is controlled by the scale of the loss and clipping.

Using the identity $\mathrm{Var}[g] = \mathbb{E}\big[\|g\|^2\big] - \|\mathbb{E}g\|^2$ and Assumption 3, we obtain

$$\mathrm{Var}[g_{\mathrm{naive}}] = \mathbb{E}\big[\|g_{\mathrm{naive}}\|^2\big] - \big\|\mathbb{E}[g_{\mathrm{naive}}]\big\|^2 \;\geq\; \mathbb{E}\big[\|g_{\mathrm{naive}}\|^2\big] - M^2, \tag{26}$$

$$\mathrm{Var}[g_{\mathrm{SIREN}}] = \mathbb{E}\big[\|g_{\mathrm{SIREN}}\|^2\big] - \big\|\mathbb{E}[g_{\mathrm{SIREN}}]\big\|^2 \;\leq\; \mathbb{E}\big[\|g_{\mathrm{SIREN}}\|^2\big]. \tag{27}$$

Combining with equation 23–equation 24, for sufficiently large $V$ we can choose a constant $C' > 1$ such that

$$\mathbb{E}\big[\|g_{\mathrm{naive}}\|^2\big] - M^2 \;\geq\; C' \,\mathbb{E}\big[\|g_{\mathrm{SIREN}}\|^2\big]. \tag{28}$$

Therefore,

$$\mathrm{Var}[g_{\mathrm{naive}}] \;\geq\; C' \,\mathbb{E}\big[\|g_{\mathrm{SIREN}}\|^2\big] \;\geq\; C' \,\mathrm{Var}[g_{\mathrm{SIREN}}], \tag{29}$$

i.e., the variance of the entropy gradient under naive entropy regularization is asymptotically larger than under SIREN, by at least a constant factor that itself grows with vocabulary size.

**Conclusion.** In the large-vocabulary, long-horizon regime, naive entropy regularization yields an entropy-gradient estimator with substantially higher variance than SIREN: noisy tail-token contributions at all timesteps inflate $\mathbb{E}\|g_{\mathrm{naive}}\|^2$, while SIREN's top-$p$ and peak-entropy masks remove most of these noise sources and keep the entropy gradient localized and low-variance.

## C EXPERIMENTAL DETAILS

### C.1 DETAILED SETUP

**Construction Method of Dataset for Llama3.1-8B.** We manually construct an easier dataset for Llama3.1-8B. Specifically, for prompts drawn from OpenR1-Math-46K-8192, the GSM8K training set, and the MATH500 training set, we employ LLaMa3.1-8B-Instruct, LLaMa3.1-72B, and LLaMa3.1-72B-Instruct to generate 8 responses per prompt. We assume that stronger models from the same series can reflect the potential performance of weaker ones. During sampling, the temperature is set to 0.6 and the maximum response length is set to 8192. We filter out prompts for which none of the models can produce a correct answer. The resulting dataset contains 35K examples. The system prompt is shown in Appendix D.

**Training Qwen2.5-Series Models.** For the Qwen2.5 series models, we sample 8 responses per prompt. The learning rate is fixed at 1e-6 with a warm-up of 5 steps. The rollout batch size is set to 128, and the mini-batch size for updates is 8 prompts, meaning that the policy is updated 16 times per rollout. The maximum response length is 3072 tokens, and the maximum prompt length is 1024 tokens; prompts exceeding this limit are filtered out before training. We train the models for 2 epochs, corresponding to a total of 714 steps.

Most experiments on Qwen2.5-Math-7B are conducted on 16 NVIDIA A800 GPUs across 2 nodes (8 GPUs per node), with two exceptions: (1) **SIREN** on Qwen2.5-Math-7B with an entropy coefficient of 0.001, and (2) the ablation experiment using only self-anchored regularization. Both are conducted on 8 NVIDIA H200 GPUs. For Qwen2.5-Math-1.5B, all experiments are conducted on 4 NVIDIA H200 GPUs.

Our implementation is based on verl (Sheng et al., 2025)[2]. We employ vLLM[3] as the rollout generator and math-verify[4] for answer extraction and verification. We gratefully acknowledge the contributions of these open-source repositories.

**Training Llama3.1-8B.** For Llama3.1-8B, due to the relative weakness of the model, we sample 16 responses per prompt. The rollout batch size is set to 512, and the mini-batch size for updates is 32, resulting in 16 policy updates per rollout. The maximum response length is 2048 tokens and the maximum prompt length is 1024 tokens. We train the model for 2 epochs on a dataset of 35K examples, corresponding to a total of 136 steps. We directly adopt the chat template of Llama3.1-8B-instruct without any modification. All experiments are conducted on 8 NVIDIA H200 GPUs.

**Validation.** For validation, we use the full test sets of the five datasets. Validation is performed every 10 steps. For each question, 16 responses are generated, with the sampling temperature set to 0.6 and the maximum response length kept consistent with training.

**Evaluation.** For evaluation, we set the temperature to 0.6 and the maximum number of new tokens to 32,768 to avoid artificially truncating responses. We use vLLM to generate the responses and math-verify to evaluate them.

## C.2 DETAILS OF BASELINE METHODS

**Dual Gradient Regularization (Haarnoja et al., 2018c).** SAC is sensitive to the choice of the temperature parameter(entropy coefficient) $\alpha$. Adjusting $\alpha$ manually is challenging because the policy entropy can fluctuate unpredictably across tasks and during training as the policy improves. To address this, an improved SAC formulation treats the problem as a constrained optimization: while maximizing the expected return, the policy is required to satisfy a minimum entropy constraint:

$$\max_{\pi} \mathbb{E}_{\pi}[R] \quad \text{s.t.} \quad H(\pi(\cdot|s)) \geq H_{\min}, \ \forall s,$$

where $H_{\min}$ is a predefined minimum policy entropy threshold.

To enforce this constraint automatically, the temperature $\alpha$ is treated as a learnable parameter and optimized via gradient ascent on the dual objective:

$$J(\alpha) = \mathbb{E}[-\alpha(H(\pi(\cdot|s_t)) - H_{\min})],$$

This allows the policy to adaptively adjust $\alpha$ during training. In our implementation, we integrate this mechanism by synchronously updating $\alpha$, with its learning rate set equal to that of the policy.

**Clip-Cov / KL-Cov (Cui et al., 2025).** This work observes that changes in policy entropy are primarily driven by the covariance between action probabilities and logit updates, which is proportional to the advantage when using policy-gradient–style updates. High-probability actions with large advantages tend to decrease entropy, whereas rare actions with large advantages increase it. This work mitigates excessive entropy decrease by suppressing updates on tokens with high covariance: Clip-Cov samples a subset of these tokens and zeros out their policy loss, while KL-Cov selects the top-k% covariance tokens and applies an additional KL penalty to constrain their update magnitude. Importantly, these methods do not compute entropy or perform entropy-based regularization; instead, they indirectly regulate entropy by controlling updates on a small set of influential tokens. As a result, they are not susceptible to entropy explosion.

**Entropy Advantage (Cheng et al., 2025).** This method applies an entropy-guided advantage shaping strategy to encourage exploratory reasoning. Concretely, it injects an entropy-dependent term $\psi(H_t) = \min(\alpha \cdot H_t^{\text{detach}}, |A_t|/\kappa)$ into the advantage function, forming $A_t^{\text{shaped}} = A_t + \psi(H_t)$. Because entropy is detached, it does not introduce gradients that directly push entropy upward. Furthermore, the shaping term is strictly bounded by $|A_t|/\kappa$, preventing runaway growth. Thus, the method emphasizes high-entropy positions while avoiding the uncontrolled positive feedback loop responsible for entropy explosion.

---

[2]https://github.com/volcengine/verl
[3]https://github.com/vllm-project/vllm
[4]https://github.com/huggingface/Math-Verify

Table 4: Overall performance on five mathematical benchmarks, extending to Qwen2.5-Math-1.5B and LLaMa3.1-8B.

| | AIME24 | | AIME25 | | AMC22 | | MATH500 | | Olympiad Bench | | Average | |
|---|---|---|---|---|---|---|---|---|---|---|---|---|
| | *Maj@32* | *Avg.* | *Maj@32* | *Avg.* | *Maj@32* | *Avg.* | *Maj@8* | *Avg.* | *Maj@8* | *Avg.* | *Maj.* | *Avg.* |
| *Qwen2.5-Math-1.5B* | | | | | | | | | | | | |
| Qwen2.5-Math-1.5B | 23.3 | 8.2 | 13.3 | 4.2 | 51.8 | 27.5 | 59.4 | 35.3 | 31.0 | 21.6 | 35.8 | 19.4 |
| Dr.GRPO | 16.7 | 14.1 | 16.7 | 8.4 | 55.4 | 44.1 | 81.2 | **77.3** | 41.5 | **37.5** | 42.3 | 36.3 |
| Naive Entropy Reg | **23.3** | **15.3** | 20.0 | 9.8 | 57.8 | **48.1** | 75.4 | 70.8 | 36.9 | 32.1 | 42.7 | 35.2 |
| *SIREN* | 20.0 | 13.8 | **23.3** | **12.0** | **60.2** | 47.9 | **80.2** | 75.7 | **41.6** | 36.0 | **45.1** | **37.0** |
| *LLaMa3.1-8B* | | | | | | | | | | | | |
| LLaMa3.1-8B | 3.3 | 0.1 | 0.0 | 0.0 | 2.4 | 1.5 | 8.8 | 5.1 | 1.5 | 1.4 | 2.5 | 1.6 |
| Dr.GRPO | 0.0 | 0.0 | 0.0 | 0.0 | 6.0 | 7.4 | 25.8 | 22.5 | 6.8 | 6.1 | 7.7 | 7.3 |
| Naive Entropy Reg | **3.3** | **0.1** | **3.3** | 0.5 | 8.4 | 7.6 | 20.6 | 19.5 | 5.6 | 5.2 | 7.6 | 6.6 |
| *SIREN* | 0.0 | 0.0 | **3.3** | **1.4** | **13.3** | **9.7** | **28.4** | **25.1** | **7.7** | **7.1** | **10.5** | **8.7** |

**RL on Forking Tokens (Wang et al., 2025).** This work identifies that only a small fraction of tokens exhibit high entropy, and these "forking" tokens critically influence downstream reasoning paths. The method improves RLVR by restricting policy-gradient updates only to these high-entropy tokens, showing that using roughly 20% of the tokens is sufficient to match or exceed full-token optimization performance. Since the method does not apply entropy regularization at all and simply filters which tokens receive policy loss, it does not introduce any mechanism that would drive entropy to diverge.

## D  SYSTEM PROMPT

> **System prompt for Qwen-series**
>
> Your task is to follow a systematic, thorough reasoning process before providing the final solution. This involves analyzing, summarizing, exploring, reassessing, and refining your thought process through multiple iterations. Structure your response into two sections: Thought and Solution. In the Thought section, present your reasoning using the format: "`<think>`\n thoughts `</think>`\n". Each thought should include detailed analysis, brainstorming, verification, and refinement of ideas. After "`</think>`\n" in the Solution section, provide the final, logical, and accurate answer, clearly derived from the exploration in the Thought section. If applicable, include the answer in `\boxed{}` for closed-form results like multiple choices or mathematical solutions.

> **System prompt for LaMa3.1-8B**
>
> Your task is to follow a systematic, thorough reasoning process before providing the final solution. This involves analyzing, summarizing, exploring, reassessing, and refining your thought process through multiple iterations. Structure your response into two sections: Thought and Solution. In the Thought section, each thought should include detailed analysis, brainstorming, verification, and refinement of ideas. In the Solution section, provide the final, logical, and accurate answer, clearly derived from the exploration in the Thought section. If applicable, include the answer in `\boxed{}` for closed-form results like multiple choices or mathematical solutions. Let's think step by step.

> **System prompt for validation and evaluation**
>
> Please reason step by step, and put your final answer within `\boxed{}`.

Table 5: Overall performance of Qwen2.5-Math-7B compared with Dr.GRPO, naïve entropy regularization, and other RLVR methods that also incorporate entropy. All models are evaluated under a unified setting. Bold indicates the best results.

| | AIME24 | | AIME25 | | AMC22 | | MATH500 | | Olympiad Bench | | Average | |
|---|---|---|---|---|---|---|---|---|---|---|---|---|
| | *Maj@32* | *Avg@32* | *Maj@32* | *Avg@32* | *Maj@32* | *Avg@32* | *Maj@8* | *Avg@32* | *Maj@8* | *Avg@32* | *Maj.* | *Avg.* |
| *SIREN* | **43.3** | 28.4 | **26.7** | **17.9** | **71.1** | 60.1 | 85.0 | **81.9** | 46.8 | 42.5 | **54.6** | **46.1** |
| w/o TopP Mask | 40.0 | 26.4 | 16.7 | 14.4 | 68.7 | **60.4** | 81.6 | 77.9 | 39.3 | 36.7 | 49.2 | 43.1 |
| w/o Peak-Entropy Mask | 33.3 | **29.0** | 26.7 | 16.8 | 65.1 | 59.2 | 80.0 | 77.9 | 39.4 | 37.3 | 48.9 | 44.0 |
| w/o Self-Anchored Reg. | 23.3 | 15.7 | 13.3 | 6.1 | 60.2 | 43.9 | 81.8 | 59.9 | 43.0 | 27.6 | 44.3 | 30.6 |
| w/ only Self-Anchored Reg. | 30.0 | 22.3 | 26.7 | 13.5 | 67.5 | 58.2 | **87.4** | 80.7 | **49.8** | **43.2** | 52.3 | 43.6 |

Table 6: Detailed experimental results of *SIREN* under different hyperparameter settings.

| | AIME24 | | AIME25 | | AMC22 | | MATH500 | | Olympiad Bench | | Average | |
|---|---|---|---|---|---|---|---|---|---|---|---|---|
| | *Maj@32* | *Avg.* | *Maj@32* | *Avg.* | *Maj@32* | *Avg.* | *Maj@8* | *Avg.* | *Maj@8* | *Avg.* | *Maj.* | *Avg.* |
| top-p=0.8, $\tau$=0.8 | **43.3** | **28.4** | **26.7** | **17.9** | **71.1** | 60.1 | 85.0 | 81.9 | 46.8 | 42.5 | **54.6** | **46.1** |
| *Varying top-p ($\tau = 0.8$)* | | | | | | | | | | | | |
| top-p=0.7 | 33.3 | 24.2 | 16.7 | 12.8 | 62.7 | 55.7 | 83.2 | 77.6 | 46.7 | 40.8 | 48.5 | 42.2 |
| top-p=0.9 | 40.0 | 25.5 | 16.7 | 13.5 | 69.9 | 60.0 | 86.4 | 81.8 | 47.3 | 43.3 | 52.0 | 44.8 |
| *Varying $\tau$ (top-p $= 0.8$)* | | | | | | | | | | | | |
| $\tau$=0.7 | 33.3 | 21.6 | 23.3 | 15.5 | **71.1** | 61.6 | 84.6 | 80.1 | 42.1 | 38.7 | 50.9 | 43.5 |
| $\tau$=0.75 | **43.3** | 28.1 | 16.7 | 13.2 | 73.5 | 62.3 | **86.6** | **82.0** | **50.4** | **44.2** | 54.1 | 46.0 |
| $\tau$=0.85 | 40.0 | 21.5 | 23.3 | 14.2 | **71.1** | **61.7** | 82.6 | 79.9 | 45.9 | 41.4 | 52.6 | 43.7 |
| $\tau$=0.9 | 26.7 | 16.0 | 23.3 | 13.1 | 69.9 | 54.9 | 84.4 | 78.3 | 47.1 | 41.1 | 50.3 | 40.7 |

# E ADDITIONAL RESULTS

## E.1 EXTENDING TO OTHER MODELS

Table 4 reports detailed results on five mathematical benchmarks. On Qwen2.5-Math-1.5B, *SIREN* achieves an average maj@k of 45.1 and avg@k of 37.0, outperforming naive baselines by +2.4 on average maj@k, respectively. On Llama3.1-8B, *SIREN* attains 10.5 maj@k and 8.7 avg@k, also surpassing all other baselines.

## E.2 ABLATION STUDY

**Hyperparameters** For the ablation experiment with the self-anchored regularization, we set the entropy coefficient to 0.005, consistent with the main experiment. For the experiment without self-anchored regularization, we set the entropy coefficient to 0.0001 to avoid entropy explosion as much as possible.

**Results** The detailed performance of the ablation study is reported in Table 5. As shown, removing each component (top-p mask, peak-entropy mask, or self-anchored regularization) leads to a performance drop, with the removal of the self-anchored regularization causing the most severe decline: -10.3 maj@k and -15.5 avg@k. This highlights the critical role of self-anchored regularization in maintaining training stability and enabling the effectiveness of other modules.

Since the performance degradation caused by removing the self-anchored regularization is too significant, we also conduct an ablation experiment using only this component. The results show that self-anchored regularization can also improve the performance of naive entropy regularization, yielding notable gains, especially on relatively easier benchmarks such as MATH500 and Olympiad-Bench. However, the overall average performance still lags behind *SIREN*, indicating that each component in *SIREN* contributes effectively.

## E.3 SENSITIVITY ANALYSIS

Table 6 presents the detailed experimental results of *SIREN* under different hyperparameter settings. We observe that when the top-p cutoff is increased or the quantile $\tau$ is increased or decreased, the

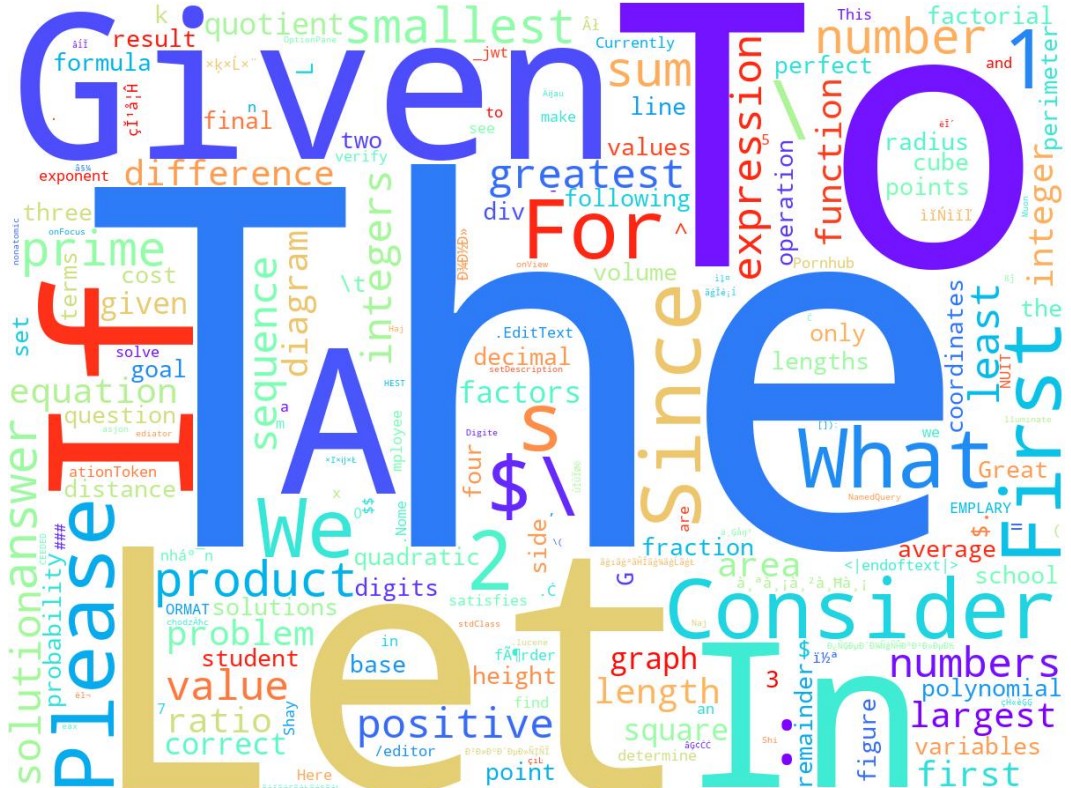

Figure 7: Word cloud of tokens selected by *SIREN*.

performance only shows a slight decline. This indicates that *SIREN* is robust and does not rely heavily on fine-tuning these hyperparameters.

## F    ANALYSIS

### F.1    *SIREN* SELECTS EFFECTIVE TOKENS

To provide a more intuitive illustration of what tokens are selected by our method, we construct a word cloud. Specifically, we randomly sample 8 questions from MATH500 and use Qwen2.5-7B-Math to generate 16 responses for each question, with the hyperparameters kept consistent with those used in the evaluation. We then collect the selected tokens and draw a word cloud with a maximum of 200 words (see Fig. 7). As shown in the figure, the high-frequency words are concentrated on core terms closely related to mathematical problem statements and reasoning, such as Given, The, To, Answer, and Consider, along with typical mathematical concepts such as product, sum, value, and equation. This demonstrates that our method effectively captures the key information in problem descriptions and solution processes rather than being dominated by redundant or noisy tokens, thereby further validating the rationality and effectiveness of the proposed mask strategy.

### F.2    *SIREN* MAKES EXPLORATION MORE EFFECTIVE (PART 2)

We study the training dynamics of the RLVR baseline, with results shown in Figure 8. Consistent with the main text, *SIREN* maintains higher entropy and pass@k, indicating more sustained exploration. Interestingly, *Kl-Cov* also preserves entropy and pass@k well; however, it introduces additional computational overhead during training due to the need to compute the KL divergence with the original model.

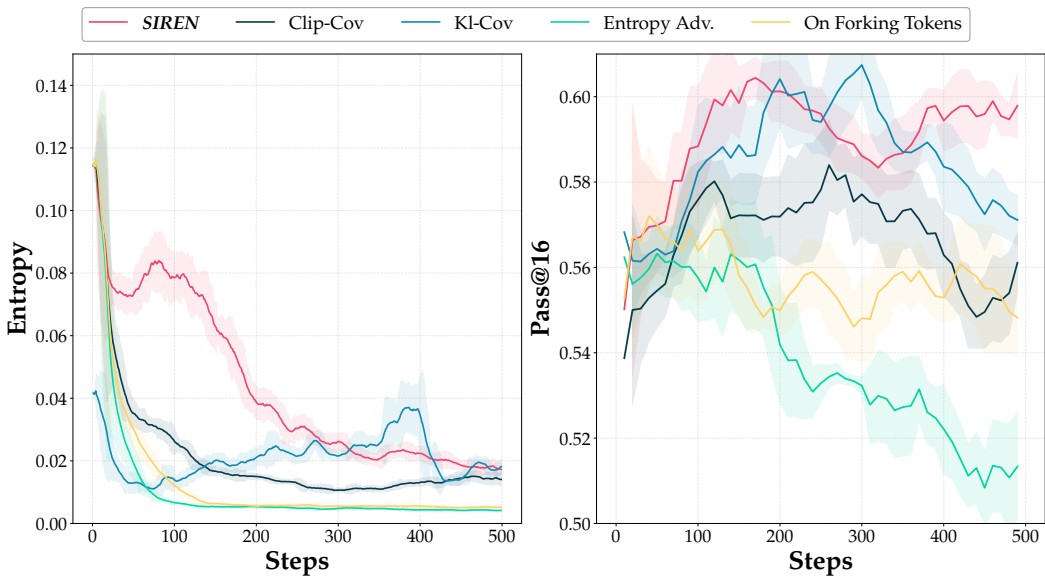

Figure 8: Training dynamics of entropy and validation pass@16, comparing **SIREN** with other RLVR baselines.

# G CASE STUDY

## G.1 MORE CASES OF PRELIMINARY ANALYSIS

We provide additional cases comparing generated tokens from the original model and the entropy-exploded model, as shown in Fig. 9. The observations are consistent with the main text: (1) after RL training with entropy explosion, the generated content becomes meaningless; (2) in the original model, tokens with higher entropy tend to correspond to logically critical positions, such as initiating or bridging a line of reasoning and elaborating individual reasoning steps, and so on.

## G.2 COMPARISON OF RESPONSES GENERATED BY **SIREN** AND RL BASELINES

To qualitatively demonstrate the improvement of the **SIREN** mechanism in structured exploration, we randomly selected two cases from the AIME24 (Li et al., 2024) task that were correctly solved by the **SIREN** model but failed by the baseline model for analysis. Compared with the baseline *Dr.GRPO*, **SIREN** produces more detailed solution steps, tends to break down the problem into finer-grained steps using more tokens, and includes a complete verification process, resulting in more accurate answers.

Example 1:

---

**Question and Ground Truth**

**Question:** Alice chooses a set $A$ of positive integers. Then Bob lists all finite nonempty sets $B$ of positive integers with the property that the maximum element of $B$ belongs to $A$. Bob's list has 2024 sets. Find the sum of the elements of A.
**Ground Truth:** 55

---

**Response from RL Baseline**                      **Total Length: 1043 Tokens**

[Problem Setup] To solve this problem, let's start by understanding the relationship between the set $A$ and the sets $B$ that Bob lists. The key point is that the maximum element of each set $B$ must be an element of $A$.

---

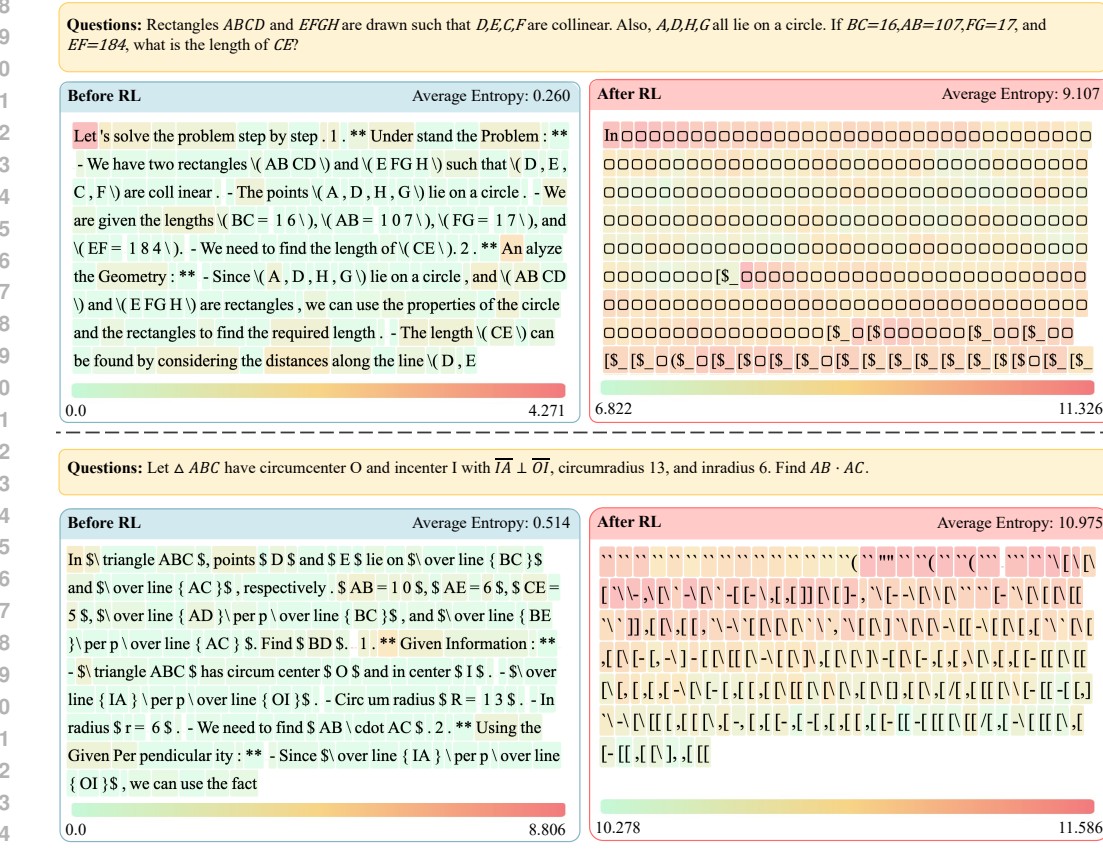

Figure 9: More comparison cases between the original model and the entropy-exploded model, with questions sampled from AIME25 (first 200 tokens).

[Strategy] If $A$ has $n$ elements, then the number of possible sets $B$ is related to the number of ways we can choose a maximum element from $A$ and then choose any subset of the elements less than the maximum element to be the remaining elements of $B$.

[Equation Transformation] If $A$ has $n$ elements, then for each element $a \in A$, we can form sets $B$ where $a$ is the maximum element. The number of such sets $B$ is $2^{a-1} - 1$ (we subtract 1 to exclude the empty set). Summing this over all elements of $A$ gives the total number of sets $B$. If $A = \{a_1, a_2, \ldots, a_n\}$, then the total number of sets $B$ is:

$$\sum_{i=1}^{n}(2^{a_i-1} - 1) = \sum_{i=1}^{n} 2^{a_i-1} - n$$

We know that this sum is equal to 2024. So we need to find a set $A$ such that:

$$\sum_{i=1}^{n} 2^{a_i-1} - n = 2024$$

[Candidate Check] This can be rearranged to:

$$\sum_{i=1}^{n} 2^{a_i-1} = 2024 + n$$

One way to approach this is to start by finding the smallest possible set $A$ and then iterating to find the correct set. Let's start by assuming $A$ has the smallest possible elements, i.e., $A =$

$\{1, 2, 3, \ldots, n\}$. Then the sum becomes:

$$\sum_{i=1}^{n} 2^{i-1} = 2^0 + 2^1 + 2^2 + \cdots + 2^{n-1} = 2^n - 1$$

We need to find $n$ such that:

$$2^n - 1 = 2024 + n$$

[Incorrect Assumption] We can solve this equation by trial and error. Let's start by checking small values of $n$. For $n = 11$:

$$2^{11} - 1 = 2047 \quad \text{and} \quad 2024 + 11 = 2035$$

This is close but not quite right. Let's try $n = 10$:

$$2^{10} - 1 = 1023 \quad \text{and} \quad 2024 + 10 = 2034$$

This is too small. So $n = 11$ is the correct value. Now we need to find the set $A$ that satisfies the equation. The set $A$ must be $\{1, 2, 3, \ldots, 11\}$, but we need to check if there is a smaller set that works.

[Incorrect Conclusion] The correct set $A$ is $\{11, 10, 9, 8, 7, 6, 5, 4, 3, 2, 1\}$. The sum of the elements of $A$ is:

$$11 + 10 + 9 + 8 + 7 + 6 + 5 + 4 + 3 + 2 + 1 = 66$$

[Incorrect Answer] So the final answer is:

$$\boxed{66}$$

---

**Response from *SIREN***            **Total Length: 1987 Tokens**

[Problem Setup] Let's begin by understanding the problem. We are given that Bob lists all finite nonempty sets $B$ of positive integers such that the maximum element of $B$ belongs to a set $A$ chosen by Alice. The number of such sets $B$ is 2024, and we need to find the sum of the elements of $A$.

[Variable Definition] Let's denote the set $A$ as $A = \{a_1, a_2, \ldots, a_n\}$ where $a_1 < a_2 < \ldots < a_n$. For each element $a_i$ in $A$, the sets $B$ that have $a_i$ as their maximum element are all the nonempty subsets of $\{1, 2, \ldots, a_i - 1\}$. The number of nonempty subsets of a set with $a_i - 1$ elements is $2^{a_i - 1} - 1$. Therefore, the total number of sets $B$ is the sum of these for all $a_i$ in $A$:

$$(2^{a_1 - 1} - 1) + (2^{a_2 - 1} - 1) + \cdots + (2^{a_n - 1} - 1) = 2024$$

[Equation Transformation] This simplifies to:

$$2^{a_1 - 1} + 2^{a_2 - 1} + \cdots + 2^{a_n - 1} - n = 2024$$

$$2^{a_1 - 1} + 2^{a_2 - 1} + \cdots + 2^{a_n - 1} = 2024 + n$$

[Strategy] We need to find a set of integers $\{a_1, a_2, \ldots, a_n\}$ such that the sum of powers of 2 is $2024 + n$. To find the correct set, we first note that $2024 + n$ must be expressible as a sum of distinct powers of 2. The largest power of 2 less than or equal to 2024 is $2^{10} = 1024$. We can start by expressing 2024 in binary:

$$2024_{10} = 11111101000_2 = 1024 + 512 + 256 + 128 + 64 + 32 + 8$$

This means:

$$2024 = 2^{10} + 2^9 + 2^8 + 2^7 + 2^6 + 2^5 + 2^3$$

[Candidate Check] If we add $n$ to 2024, we need to check if the resulting sum can be expressed as a sum of distinct powers of 2. To find the correct $n$, we can check the next smallest values. If $n = 1$, then:

$$2024 + 1 = 2025$$

The binary representation of 2025 is:

$$2025_{10} = 11111101001_2 = 1024 + 512 + 256 + 128 + 64 + 32 + 8 + 1$$

This means:

$$2025 = 2^{10} + 2^9 + 2^8 + 2^7 + 2^6 + 2^5 + 2^3 + 2^0$$

This is a sum of distinct powers of 2, so it is valid. Therefore, the set $A$ could be $\{11, 10, 9, 8, 7, 6, 3, 1\}$. The sum of the elements of $A$ is:

$$11 + 10 + 9 + 8 + 7 + 6 + 3 + 1 = 55$$

[Verification] To verify, the number of sets $B$ is:

$$(2^{11-1}-1)+(2^{10-1}-1)+(2^{9-1}-1)+(2^{8-1}-1)+(2^{7-1}-1)+(2^{6-1}-1)+(2^{3-1}-1)+(2^{1-1}-1)$$

$$= (2^{10} - 1) + (2^9 - 1) + (2^8 - 1) + (2^7 - 1) + (2^6 - 1) + (2^5 - 1) + (2^2 - 1) + (2^0 - 1)$$

$$= 1024 - 1 + 512 - 1 + 256 - 1 + 128 - 1 + 64 - 1 + 32 - 1 + 4 - 1 + 1 - 1$$

$$= 1024 + 512 + 256 + 128 + 64 + 32 + 4 + 1 - 8$$

$$= 2025 - 8 = 2017$$

This is not 2024, so we need to check the next value. If $n = 2$, then:

$$2024 + 2 = 2026$$

The binary representation of 2026 is:

$$2026_{10} = 11111101010_2 = 1024 + 512 + 256 + 128 + 64 + 32 + 8 + 2$$

This means:

$$2026 = 2^{10} + 2^9 + 2^8 + 2^7 + 2^6 + 2^5 + 2^3 + 2^1$$

This is a sum of distinct powers of 2, so it is valid. Therefore, the set $A$ could be $\{11, 10, 9, 8, 7, 6, 3, 1\}$ with an additional element, say 2. The sum of the elements of $A$ is:

$$11 + 10 + 9 + 8 + 7 + 6 + 3 + 1 + 2 = 57$$

But, the number of sets $B$ is:

$$(2^{11-1}-1)+(2^{10-1}-1)+(2^{9-1}-1)+(2^{8-1}-1)+(2^{7-1}-1)+(2^{6-1}-1)+(2^{3-1}-1)+(2^{1-1}-1)$$
$$+(2^{2-1} - 1)$$

$$= (2^{10}-1)+(2^9-1)+(2^8-1)+(2^7-1)+(2^6-1)+(2^5-1)+(2^2-1)+(2^0-1)+(2^1-1)$$

$$= 1024 - 1 + 512 - 1 + 256 - 1 + 128 - 1 + 64 - 1 + 32 - 1 + 4 - 1 + 1 - 1 + 2 - 1$$

$$= 1024 + 512 + 256 + 128 + 64 + 32 + 4 + 1 + 2 - 9$$

$$= 2026 - 9 = 2017$$

This is not 2024, so the correct set $A$ must be $\{11, 10, 9, 8, 7, 6, 3, 1\}$ with $n = 1$. The sum of the elements of $A$ is 55. So, the final answer is:

[Conclusion]

$$\boxed{55}$$

Example 2:

---

**Question and Ground Truth**

**Question:** Let $\omega \neq 1$ be a 13th root of unity. Find the remainder when

$$\prod_{k=0}^{12}(2 - 2\omega^k + \omega^{2k})$$

is divided by 1000.
**Ground Truth:** 321

---

**Response from RL Baseline**            **Total Length: 808 Tokens**

[Problem Setup and Polynomial Definition] To solve the problem, we need to evaluate the product $\prod_{k=0}^{12}(2 - 2\omega^k + \omega^{2k})$ where $\omega$ is a 13th root of unity, and then find the remainder when the result is divided by 1000.
First, let's understand the expression $2 - 2\omega^k + \omega^{2k}$. Since $\omega$ is a 13th root of unity, $\omega^{13} = 1$ and $\omega^k$ for $k = 0, 1, 2, \ldots, 12$ are all distinct 13th roots of unity. We need to evaluate the product for $k = 0, 1, 2, \ldots, 12$.
Let's denote the polynomial $P(x) = 2 - 2x + x^2$. We need to evaluate the product $P(\omega^0) \cdot P(\omega^1) \cdot P(\omega^2) \cdot \ldots \cdot P(\omega^{12})$.
First, we evaluate $P(\omega^k)$:

$$P(\omega^k) = 2 - 2\omega^k + (\omega^k)^2 = 2 - 2\omega^k + \omega^{2k}.$$

Now, we need to find the product of these values for $k = 0, 1, 2, \ldots, 12$.
[Flawed Mathematical Reasoning/Incorrect Formula Application] To simplify the problem, we can use the fact that the polynomial $P(x) = 2 - 2x + x^2$ can be rewritten in a more convenient form. Notice that:

$$P(x) = x^2 - 2x + 2.$$

We need to evaluate the product:

$$\prod_{k=0}^{12}(2 - 2\omega^k + \omega^{2k}).$$

This product can be evaluated by considering the polynomial $P(x)$ evaluated at the 13th roots of unity. The polynomial $P(x)$ can be written as:

$$P(x) = x^2 - 2x + 2.$$

We need to find the value of the product of $P(\omega^k)$ for $k = 0, 1, 2, \ldots, 12$. This can be done by considering the polynomial $P(x)$ and its roots.
[Incorrect Resultant/Vieta's Formula Misapplication] The polynomial $P(x)$ can be evaluated at the roots of unity. The product of the values of a polynomial at the roots of unity can be found using the fact that the product of the values of a polynomial $P(x)$ at the roots of unity is given by the constant term of the polynomial $P(x)$ raised to the power of the number of roots, divided by the leading coefficient raised to the power of the number of roots minus one.
In this case, the polynomial $P(x) = x^2 - 2x + 2$ has a constant term of 2 and a leading coefficient of 1. The product of the values of $P(x)$ at the 13th roots of unity is given by $2^{13}/1^{12} = 2^{13} = 8192$.
[Modulo Operation and Incorrect Final Answer] Finally, we need to find the remainder when 8192 is divided by 1000. This is given by:

$$8192 \mod 1000 = 192.$$

So the remainder when the product is divided by 1000 is $\boxed{192}$.

**Response from *SIREN***                                          **Total Length: 1417 Tokens**

[Problem Setup and Initial Transformation] To solve this problem, we need to consider the expression $\prod_{k=0}^{12}(2 - 2\omega^k + \omega^{2k})$, where $\omega$ is a 13th root of unity, i.e., $\omega^{13} = 1$ and $\omega \neq 1$. Let's first rewrite the expression inside the product:

$$2 - 2\omega^k + \omega^{2k} = 2(1 - \omega^k) + \omega^{2k}.$$

[Contextual Background - Roots of Unity] Next, we will look at the polynomial whose roots are $\omega^k$ for $k = 0, 1, 2, \ldots, 12$. The polynomial is $x^{13} - 1 = 0$, which can be factored as:

$$x^{13} - 1 = (x - 1)(x - \omega)(x - \omega^2) \cdots (x - \omega^{12}).$$

[Reframing the Problem as Polynomial Product] We need to find the value of the product $\prod_{k=0}^{12}(2 - 2\omega^k + \omega^{2k})$. We can consider the polynomial $P(x) = 2 - 2x + x^2$, and we need to find the product $P(\omega^0) \cdot P(\omega^1) \cdot P(\omega^2) \cdots P(\omega^{12})$.

[Attempted Solution - First Approach (Incorrect Formula)] To find this product, we can consider the polynomial whose roots are $P(\omega^k)$ for $k = 0, 1, 2, \ldots, 12$. Let's define a new polynomial $Q(y)$ such that the roots of $Q(y)$ are $P(\omega^k)$. We can find $Q(y)$ by considering the transformation $y = P(x) = 2 - 2x + x^2$. To find $Q(y)$, we need to express $x$ in terms of $y$. We have:

$$x^2 - 2x + 2 - y = 0.$$

The roots of this quadratic equation are:

$$x = 1 \pm \sqrt{y - 1}.$$

Since $\omega^k$ are the 13th roots of unity, we need to find the polynomial whose roots are $P(\omega^k)$. The polynomial $Q(y)$ will be the resultant of the polynomial $x^{13} - 1$ and the transformation $y = 2 - 2x + x^2$. However, a simpler approach is to note that the product of the values of a polynomial at the roots of unity is given by the resultant of the polynomial and $x^n - 1$, where $n$ is the degree of the polynomial. In this case, we can use the fact that the product of $P(\omega^k)$ for $k = 0, 1, 2, \ldots, 12$ is the value of the polynomial $P(x)$ raised to the power of the degree of the polynomial $x^{13} - 1$, evaluated at the roots of unity.

[Attempted Solution - Second Approach (Incorrect/Incomplete Formula)] The polynomial $P(x) = 2 - 2x + x^2$ can be written as $P(x) = (x - 1)^2 + 1$. The product of $P(\omega^k)$ for $k = 0, 1, 2, \ldots, 12$ is the value of the polynomial $P(x)$ evaluated at the roots of unity, which is the constant term of the polynomial $P(x)^{13}$ divided by the leading coefficient. This is given by:

$$\prod_{k=0}^{12} P(\omega^k) = P(1)^{13} = (2 - 2 \cdot 1 + 1^2)^{13} = 1^{13} = 1.$$

[Self-Correction and Recognition of Complexity] However, this is not the correct approach for this specific problem. We need to consider the polynomial whose roots are $2 - 2\omega^k + \omega^{2k}$. Let's define a new polynomial $R(x)$ such that the roots of $R(x)$ are $2 - 2\omega^k + \omega^{2k}$. We can find $R(x)$ by considering the transformation $x = 2 - 2\omega^k + \omega^{2k}$. To find $R(x)$, we need to express $\omega^k$ in terms of $x$. However, a simpler approach is to note that the product of the values of a polynomial at the roots of unity is given by the resultant of the polynomial and $x^n - 1$, where $n$ is the degree of the polynomial. In this case, we can use the fact that the product of $2 - 2\omega^k + \omega^{2k}$ for $k = 0, 1, 2, \ldots, 12$ is the value of the polynomial $P(x)$ raised to the power of the degree of the polynomial $x^{13} - 1$, evaluated at the roots of unity.

[Pragmatic Computational Approach] Let's use Python to calculate the product $\prod_{k=0}^{12}(2 - 2\omega^k + \omega^{2k})$ and find the remainder when this product is divided by 1000.

```
import numpy as np

# 13th roots of unity
omega = np.exp(2j * np.pi / 13)
roots_of_unity = [omega**k for k in range(13)]

# Calculate the product
```

```
product = np.prod([2 - 2*omega**k + omega**(2*k) for k in
range(13)])

# Get the real part of the product (since the imaginary part
b should be very close to 0)
product_real = np.real(product)

# Find the remainder when the product is divided by 1000
remainder = int(round(product_real)) % 1000
print(remainder)

321
```

[Final Answer] The remainder when $\prod_{k=0}^{12}(2 - 2\omega^k + \omega^{2k})$ is divided by 1000 is $\boxed{321}$.

