# OpenReview forum: "Rethinking Entropy Regularization in Large Reasoning Models"
_ICLR.cc/2026/Conference — Submitted to ICLR 2026_

### Official Review · Reviewer_ys5j · 2025-10-14

**Soundness:** 3
**Presentation:** 3
**Contribution:** 2
**Rating:** 2
**Confidence:** 3

**Summary:**

The paper introduces SIREN (SelectIve entRopy rEgularizatioN), an entropy-regularization variant for reinforcement learning with verifier rewards (RLVR). The key idea is to selectively apply entropy regularization via (1) a top-p mask restricting exploration to high-probability “policy nucleus” tokens, (2) a peak-entropy mask focusing on high-entropy positions along reasoning trajectories, and (3) a self-anchored regularization term keeping global entropy close to its initial value. Experiments on math reasoning benchmarks (AIME 24/25, AMC, OlympiadBench, MATH500) with Qwen2.5-Math-7B/1.5B and LLaMA 3.1-8B show moderate performance gains (+4.8 maj@k on average vs. Dr.GRPO) and more stable training dynamics.

**Strengths:**

- Timely topic – Addresses the practical issue of entropy collapse and premature convergence in RLVR for large reasoning models.

- Empirical clarity – Provides informative analyses (entropy distributions, trajectory-level entropy heatmaps) diagnosing “entropy explosion” with naive regularization.

**Weaknesses:**

## Limited Novelty

- Top-p masking: Using top-p sampling for language models is well-established (Holtzman et al., 2019, cited by authors). Applying it to entropy computation is a straightforward extension, not a fundamental insight.

- Self-anchored regularization: Computing regularization loss as MSE from an initial anchor is a relatively simple stabilization technique. The novelty lies mainly in the combination rather than individual components.

- Peak-entropy masking: Identifying important tokens via entropy is also explored by concurrent work (Wang et al., 2025; Cheng et al., 2025). The quantile-based selection (Eq. 9) is reasonable but incremental. These are all very recent papers (May-July 2025 based on arXiv dates), which explains why they're labeled "concurrent work"—they were likely being developed around the same time as SIREN.

1. Beyond the 80/20 rule: High-entropy minority tokens drive effective reinforcement learning for llm reasoning.

2. Reasoning with exploration: An entropy perspective.

3. The entropy mechanism of reinforcement learning for reasoning language models.



## Lack of theoretical grounding

The paper repeatedly claims to “rethink entropy regularization”, yet provides no formal analysis or theoretical justification for why the proposed masking mechanism should mitigate explosion or collapse beyond intuition and empirical plots. There is no attempt to connect the approach to established RL theory (e.g., maximum-entropy RL, policy improvement bounds, or KL-controlled exploration). This weakens the “rethinking” claim.

## Framing and writing inflation

The title and abstract strongly over-sell the contribution (“rethinking entropy regularization”, “foundational insight”), whereas the actual contribution is heuristic and incremental. Much of Section 2–3 reiterates known observations about entropy explosion without introducing new analysis tools.

**Questions:**

- How sensitive are results to p (top-p cutoff) and τ (peak-entropy quantile)?

- Does SIREN generalize to domains beyond math, e.g., code generation or open-ended reasoning?

- Can the authors compare to methods like KL-controlled adaptive entropy (e.g., SAC-style) or dual-gradient regularization rather than hand-crafted masks?

- Can the authors investigate whether there are improvements over Pass@k (a larger k)? Several recent papers have studied the reasoning boundaries on math and other benchmarks.

1. Does reinforcement learning really incentivize reasoning capacity in llms beyond the base model?

2. The invisible leash: Why rlvr may or may not escape its origin?

---

> ### Author Response · Authors · 2025-11-21
> **Summary of Response**
>
> We sincerely thank Reviewer ys5j for the insightful comments. Your feedback has motivated us to provide additional analyses to clarify the motivation behind SIREN and to explain the potential reasons for its effectiveness. We address your comments as follows:
>
> - Motivation and effectiveness of SIREN.
> - The theoretical grounding of SIREN.
> - How we rethink entropy regularization in LRMs, and a new candidate title.
> - SIREN’s generalizability to domains beyond mathematics.
> - Comparison with SAC-style entropy regularization.
> - SIREN’s improvements in Pass@k across datasets.
>
> Regarding the newly introduced hyperparameters, **topP cutoff** and **quantile**, we are currently running sensitivity analyses and will report the results during the rebuttal period. We hope that our responses address your concerns.

---

> > ### Author Response · Authors · 2025-11-21
> > **The theoretical grounding of SIREN.**
> >
> > > Q: The paper repeatedly claims to “rethink entropy regularization”, yet provides no formal analysis or theoretical justification for why the proposed masking mechanism should mitigate explosion or collapse beyond intuition and empirical plots. There is no attempt to connect the approach to established RL theory (e.g., maximum-entropy RL, policy improvement bounds, or KL-controlled exploration). This weakens the “rethinking” claim.
> > >
> >
> > We thank the reviewer for raising this important point. To strengthen the theoretical grounding of SIREN, we have added a detailed analysis in appendix B.  Specifically, we show that naive entropy regularization in LRMs is either unstable, because it is prone to entropy explosion, or ineffective, because it provides negligible exploration incentives. In contrast, SIREN’s masking design explicitly **bounds the entropy gradients**, preventing unstable updates and ensuring that the entropy signal remains meaningful. Moreover, the masking mechanisms substantially **reduce the variance of entropy gradients**, which stabilizes optimization.
> >
> > Regarding the self-anchored regularization term, we further clarify its theoretical motivation: similar to the principle behind SAC, it encourages entropy to converge toward a **target range** rather than increasing unboundedly.

---

> ### Author Response · Authors · 2025-11-21
> **Motivation and effectiveness of SIREN.**
>
> > Q:  Limited Novelty
>
> We thank the reviewer for the question, and we address the effectiveness and contribution of SIREN from the following perspectives.
>
> 1. **Clear motivation rooted in a concrete failure mode.**
>
> We diagnose that naive entropy regularization in LRMs often triggers *entropy explosion*. Through analytical reasoning and empirical observations, we find that this failure mode arises from the **vast action space** and **long state trajectories** of LRMs: Actions with extremely small probabilities, which are often semantically irrelevant, are inadvertently encouraged, leading the action distribution to become overly uniform and diffuse throughout the sequence.
>
> To directly address this issue, we introduce **top-p masking** and **peak-entropy masking** to *bound* the effective entropy signal, and **self-anchored regularization** to *control its magnitude* and stabilize training.
>
> These components are designed as targeted solutions to the diagnosed failure mode; therefore, the overall method is **principled rather than arbitrary or ad-hoc**.
>
> 1. **Simple yet consistently effective.**
>
> Previous work, such as SimPO[1], s1[2], has been accepted despite using simple yet effective methods, demonstrating that simplicity alone does not preclude acceptance. Similarly, SIREN, although lightweight, directly constrains unstable entropy dynamics, yielding substantial improvements over prior entropy-based RL methods. Ablation studies further confirm that removing any single component significantly degrades performance, indicating that SIREN’s effectiveness stems from addressing the underlying structural issue rather than relying on superficial heuristics.
>
> [1] SimPO: Simple Preference Optimization with a Reference-Free Reward
>
> [2] s1: Simple test-time scaling

---

> > ### Comment · Reviewer_ys5j · 2025-11-21
> >
> > Thanks for your clarification on your motivation. However, I am still concerned about the entropy regularization setting. Notably, there are several other ways to prevent the entropy collapse. For instance, DAPO proposed the clipping-high strategy, which raised the ceiling of PPO. Have you discussed and compared your algorithm with theirs?
> >
> >
> > Moreover, I also agree that simple but effective approaches are elegant and should be accepted into the top-tier conferences. However, you need to clarify your distinction from those prior studies, as I listed previously. You picked up SimPO/s1 as evidence, which cannot support the novelty of this paper.

---

> > > ### Author Response · Authors · 2025-11-23
> > >
> > > We sincerely thank the reviewer for the thoughtful comments and for carefully examining our work. Your feedback helps us clarify the motivation and contributions of SIREN more clearly.
> > >
> > > ### **Relation to DAPO/clip-higher**
> > >
> > > Regarding DAPO/clip-higher, our experiments already adopt this strategy: we use a **clip ratio of 0.28 instead of the default 0.2**, effectively placing all baselines under a raised PPO ceiling. SIREN is therefore not an alternative to clip-higher, but a complementary mechanism that further stabilizes exploration by directly refining entropy regularization.
> > >
> > > ---
> > >
> > > ### **Distinction from Prior Studies**
> > >
> > > SIREN is motivated by a fundamentally different principle. Existing RLVR methods that utilize entropy fall into two conceptual categories:
> > >
> > > **(1) Entropy-value–based methods** (e.g., *entropy-adv*, *on-forking-tokens*).
> > >
> > > These approaches detach entropy gradients and use only the **entropy value** as an intrinsic signal. Conceptually, they are all performing **credit assignment** on top of the original advantage:
> > >
> > > - *entropy-adv* adds the entropy value to the advantage, thereby amplifying the credit of high-entropy tokens;
> > > - *on-forking-tokens* updates the policy only on high-entropy tokens, which is effectively another form of credit assignment—tokens with low entropy have their advantages set to zero.
> > >
> > > SIREN shares with this category the high-level idea of leveraging entropy, but differs fundamentally in *purpose*:
> > >
> > > Entropy-value methods **do not include an entropy regularization term**; instead, they perform credit assignment using entropy values. In contrast, **SIREN is an optimization of entropy regularization** itself. Rather than detaching entropy gradients, SIREN restricts the scope of entropy regularization so that policy entropy remains controlled and exploration stays effective throughout training.
> > >
> > > This difference is also reflected empirically in Figure 8: SIREN preserves stable entropy dynamics and yields consistent gains in validation pass@k, a pattern not observed in entropy-value approaches
> > >
> > > **(2) Covariance-based methods** (e.g., *clip-cov*, *kl-cov*).
> > >
> > > These methods start from the empirical observation that entropy reduction is primarily driven by the **covariance between action probabilities and logit updates**, and therefore attempt to maintain exploration **indirectly** by suppressing high-covariance token updates. SIREN is related to this line of work in that we share the same motivation: *preventing rapid entropy collapse to avoid early policy convergence*.
> > >
> > > However, the approaches diverge fundamentally in how this goal is pursued.
> > >
> > > Because *clip-cov* and *kl-cov* observe the ineffectiveness of naive entropy regularization, they move away from the entropy-regularization route entirely, and these methods do not even compute entropy, and instead aim to control exploration by restricting updates on selected tokens.
> > >
> > > By contrast, SIREN takes the opposite stance when confronted with the same failure mode. Rather than abandoning entropy regularization, SIREN **analyzes why the naive form breaks down** and introduces a lightweight confinement mechanism that makes entropy regularization stable and functional. In this way, SIREN preserves the original conceptual advantages of entropy regularization while providing the controllability and robustness that prior baselines sought through indirect means.
> > >
> > > We hope that the above clarifications help elucidate the conceptual differences between SIREN and prior studies, and clearly highlight where its novelty lies.

---

> ### Author Response · Authors · 2025-11-21
> **How we rethink entropy regularization in LRMs, and a new candidate title.**
>
> > Q: a Framing and writing inflation: The title and abstract strongly over-sell the contribution (“rethinking entropy regularization”, “foundational insight”), whereas the actual contribution is heuristic and incremental. Much of Section 2–3 reiterates known observations about entropy explosion without introducing new analysis tools.
> >
>
> We thank the reviewer for the thoughtful comment. We would like to clarify that Sections 2–3 do not merely restate previously known observations about entropy explosion. Beyond reporting empirical phenomena, we provide a **new analytical explanation** of why LRMs are particularly vulnerable to entropy explosion. Specifically, by analyzing the entropy regularization formula, we show that the instability arises from the combination of (1) the vast action space and (2) the long trajectory length characteristic of LRMs—two factors that amplify small-probability actions and push the distribution toward uniformity. We further support this claim with **sequence-level and token-level visualizations** that reveal how action probabilities evolve during explosion, providing evidence not discussed in prior work. Moreover, as elaborated in A2, we include a **theoretical analysis in the appendix B** that formally justifies our conclusions. We hope this addresses the reviewer’s concerns.
>
> At the same time, we appreciate the reviewer’s concern regarding the framing of our contribution. Your comment is constructive, and we agree that the title and abstract could more precisely reflect the nature of the work. We are open to refining the framing and would be happy to adjust the title accordingly. For example, we could consider a more focused and modest alternative such as
>
> **“On Effective Entropy Regularization for Large Reasoning Models.”**
>
> We would greatly value the reviewer’s opinion on whether this framing better captures the contribution.
>
> We hope this addresses the reviewer’s concerns.

---

> ### Author Response · Authors · 2025-11-21
> **SIREN’s generalizability to domains beyond mathematics.**
>
> > Q: Does SIREN generalize to domains beyond math, e.g., code generation or open-ended reasoning
> >
>
> We thank the reviewer for the question. We acknowledge that our experiments are currently limited to the math reasoning domain. Due to computational and time constraints, we were unable to extend the evaluation to additional domains such as code generation or open-ended reasoning. We also note that several of our baselines (e.g., clip cov, entropy advantage, on forking tokens) similarly report results only in the math setting.
>
> That said, naive entropy regularization is widely used across diverse reasoning tasks, and SIREN directly improves this core mechanism. Therefore, we believe that SIREN has strong potential to generalize to other domains. We consider this a direction for future work.

---

> ### Author Response · Authors · 2025-11-21
> **Comparison with SAC-style entropy regularization.**
>
> > Q:  Can the authors compare to methods like KL-controlled adaptive entropy (e.g., SAC-style) or dual-gradient regularization rather than hand-crafted masks?
> >
>
> We thank the reviewer for the valuable suggestion. Following this comment, we have added comparisons with **dual-gradient regularization**. In our experiments, we set the initial coefficient to 0.001 and used the same learning rate as the policy optimizer.
>
> We observe that dual-gradient regularization yields clear improvements over naive regularization, but **it still underperforms SIREN**. While SAC-style methods can make entropy more controllable by adapting the entropy coefficient online, they introduce additional tuning complexity: the initial entropy coefficient and its learning-rate require careful, often brittle, adjustment. In contrast, SIREN only requires driving policy entropy toward a target value, which substantially reduces the number of tunable hyperparameters and the sensitivity of training to their settings.
>
> |  | AIME24 |  | AIME25 |  | AMC |  | MATH500 |  | OlympiadBench |  | Average |  |
> | --- | --- | --- | --- | --- | --- | --- | --- | --- | --- | --- | --- | --- |
> |  | Maj@32 | Avg@32 | Maj@32 | Avg@32 | Maj@32 | Avg@32 | Maj@8 | Avg@8 | Maj@8 | Avg@8 | Maj@k | Avg@k |
> | Dual Gradient Reg. | 36.7 | **30.2** | 20.0 | 16.7 | 68.7 | **63.7** | 82.8 | 79.6 | 43.7 | 40.7 | 50.4 | **46.2** |
> | SIREN | **43.3** | 28.4 | **26.7** | **17.9** | **71.1** | 60.1 | **85.0** | **81.9** | **46.8** | **42.5** | **54.6** | 46.1 |
>
> ---

---

> ### Author Response · Authors · 2025-11-21
> **SIREN’s improvements in Pass@k across datasets.**
>
> > Q: Can the authors investigate whether there are improvements over Pass@k (a larger k)? Several recent papers have studied the reasoning boundaries on math and other benchmarks.
> >
>
> We have examined the behavior of **Pass@k** in detail. In Section 5.2, we study how SIREN performs as k increases, and we observe that SIREN achieves consistently high Pass@k on most datasets. This indicates that SIREN effectively **enhances exploration** and improves the **upper bound of achievable reasoning performance**.

---

> > ### Comment · Reviewer_ys5j · 2025-11-21
> >
> > Thanks for adding the pass@k experimental results. However, this looks confusing to me. For instance, why are metrics inconsistent? Some used pass@8 while others used pass@32.
> >
> > More importantly, according to your new results over pass@k, SIREN does not outperform the baseline algorithm (Dual Gradient Reg.) significantly. Instead, it was worse in some datasets and slightly worse on average @k. This makes me worry about the effectiveness of SIREN. Therefore, your statement of "improves the upper bound of achievable reasoning performance" is incorrect.

---

> > > ### Author Response · Authors · 2025-11-23
> > >
> > > Thank you very much for the thoughtful questions and for carefully examining the new pass@k results.
> > >
> > > ### **Explanation of Different k Values**
> > >
> > > The choice of k varies depending on the size and difficulty of each dataset. We follow the settings in [1] and set k=32 for AIME24, AIME25, and AMC due to their relatively small size, while for larger datasets we set k=8 instead of k=1 as in [1] to better evaluate the model’s exploration capability. When evaluating pass@k, we follow [2] and choose k according to dataset difficulty: for the more challenging datasets AIME25 and AIME24, we set k=256; for the easier dataset Math500, we set k=16; and for other datasets of intermediate difficulty, we use k=128.
> > >
> > > ---
> > >
> > > ### **Comparison with Dual Gradient Regularization**
> > >
> > > Regarding the comparison with Dual Graident Reg., we would like to emphasize that SIREN consistently achieves **higher maj@k across all datasets**. The metric maj@k selects the most frequent answer among k samples and then checks its correctness. We believe this is the most appropriate metric for evaluating **effective exploration**:
> > >
> > > (1) compared to pass@1, it reduces randomness by capturing the breadth of the model’s exploratory space across multiple generations;
> > >
> > > (2) compared to pass@k, it avoids false positives that arise from randomly guessing the correct answer, since the model must achieve a **high-confidence** correct response to score well.
> > >
> > > For these reasons, we adopt maj@k as our primary metric.
> > >
> > > From the perspective of maj@k, SIREN demonstrates more effective exploration than Dual Graident Reg., with particularly notable improvements on harder datasets such as AIME24 and AIME25. Although SIREN shows a slight reduction in avg@k, we conduct a detailed analysis on AMC, which is the dataset where the drop is most pronounced. We find that, compared with Dual Gradient Reg., SIREN produces fewer “all-correct” cases (18 out of 83, versus 29 out of 83 for Dual Gradient Reg.), but a substantially larger number of partially correct responses (32 cases with at least half of the answers correct, compared with 24 for Dual Gradient Reg.).
> > >
> > > In other words, SIREN reshapes the distribution of correctness counts: instead of concentrating on extreme outcomes (all-correct or all-wrong), it produces more intermediate cases. This behavior explains the slight decrease in avg@k, while still achieving higher maj@k, and therefore does not indicate that SIREN is ineffective, it reflects the intended effect of encouraging a broader and more balanced exploration profile.
> > >
> > > We sincerely thank the reviewer for their time and constructive feedback. We will include a more detailed analysis of these metrics and distributional effects in the revised version of the paper.
> > >
> > > [1] Learning to Reason under Off-Policy Guidance.
> > >
> > > [2] Reasoning with Exploration: An Entropy Perspective

---

> > > > ### Comment · Reviewer_ys5j · 2025-11-23
> > > >
> > > > I appreciate the quick response. However, maj@k is not the most used metric in evaluating the reasoning boundary. I have asked for pass@k (how a model can answer at least one time correctly in k attempts) instead of maj@k. Moreover, avg@k is usually more important than maj@k.
> > > >
> > > > - If the model frequently outputs a wrong answer, maj@k will confidently select it.
> > > >
> > > > - maj@k does not protect against systematic hallucination.
> > > >
> > > > - The claim that it avoids false positives relative to pass@k may appear unproven or overstated.
> > > >
> > > > - maj@k measures the most common answer, which reflects mode dominance, not exploration.
> > > >
> > > > - A model could produce many similar wrong answers → high maj@k but poor exploration.
> > > >
> > > > - Exploration should measure coverage and diversity, not just the majority vote.
> > > >
> > > > "The choice of k varies depending on the size and difficulty of each dataset." --- This does not sound rational to me. A higher k is usually better.
> > > >
> > > >
> > > > Please take a look at those papers that I previously listed, which used pass@k as the primary metric. I believe your paper is not to challenge the standard metric used in this area, but to demonstrate the expressiveness of your proposed algorithm. It is suspicious to avoid reporting pass k.
> > > >
> > > > 1. Does reinforcement learning really incentivize reasoning capacity in llms beyond the base model?
> > > >
> > > > 2. The invisible leash: Why rlvr may or may not escape its origin?

---

> > > > > ### Author Response · Authors · 2025-11-27
> > > > >
> > > > > Thank you very much for the thoughtful comment and for highlighting the importance of *pass@k* in evaluating the reasoning boundary of LLMs. We fully agree that *pass@k* is widely used and remains one of the most meaningful metrics in this line of work. We would also like to clarify that **we never intended to avoid reporting *pass@k***. In fact, **Section 5.2 and Figure 5** already present *pass@k* across a range of *k*, where we observe that **SIREN consistently achieves higher pass@k than the baseline across most datasets and values of k**. We believe this indicates that SIREN indeed improves the reasoning boundary to a certain extent.
> > > > >
> > > > > To make the results clearer, we additionally summarize the results from Figure 5 in the tabular form below. In particular:
> > > > >
> > > > > - We include results for the baseline **Dual Gradient Reg.**, as requested.
> > > > > - To directly address the reviewer’s concern, we also **re-evaluate MATH500 with k = 128**, matching the evaluation protocol used for AMC and OlympiadBench. Due to the computational constraints of the rebuttal phase, we are only able to re-run the evaluation with *k = 128* for the larger dataset. In future revisions, we will consider synchronizing all datasets to *k = 256* for completeness.
> > > > >
> > > > > |  | AIME24 | AIME25 | AMC | MATH500 | OlympiadBench |
> > > > > | --- | --- | --- | --- | --- | --- |
> > > > > |  | Pass@256 | Pass@256 | Pass@128 | Pass@128 | Pass@128 |
> > > > > | Dr.GRPO | **80** | 56.7 | 90.4 | 94.6 | 72.4 |
> > > > > | Naive Entropy Reg. | 66.7 | 53.3 | 94.0 | 96.2 | 70.7 |
> > > > > | Clip-Cov | 66.7 | 56.7 | **95.2** | 92.8 | 69.5 |
> > > > > | KL-Cov | 70 | **60** | **95.2** | 95.8 | 72 |
> > > > > | Entropy Adv. | 66.7 | 56.7 | 90.4 | 94 | 67.6 |
> > > > > | On Forking Tokens | **80** | **60** | 94.0 | 96 | 73.0 |
> > > > > | [NEW] Dual Gradient Reg. | 73.3 | 50 | 92.8 | 95.6 | 69.6 |
> > > > > |  |  |  |  |  |  |
> > > > > | SIREN | 70 | **60** | **95.2** | **97.4** | **74.1** |
> > > > >
> > > > > If the reviewer finds this helpful, we would be happy to incorporate this table into the main paper.
> > > > >
> > > > > Finally, we would like to restate the motivation for including *maj@k* (in addition to *avg@k*) in the main experiments. Our intention is not to replace *pass@k*, but to complement it with a metric that offers certain practical advantages: *maj@k* **reduces the randomness** inherent in *pass@1* by aggregating multiple samples, and, importantly, it **does not require a golden verifier**, enabling it to function as a practical test-time scaling method in scenarios where verifiers may not be available.
> > > > >
> > > > > We hope this additional analysis clarifies our intention and resolves the concern regarding evaluation choices. Once again, we appreciate your constructive feedback, which helped us improve the clarity of our presentation.

---

> > > > > > ### Comment · Reviewer_ys5j · 2025-11-27
> > > > > >
> > > > > > Hi, thanks for your great efforts in answering my question. This new table looks much better with additional pass k evaluation. However, it seemed that SIREN does not outperform the baselines in some datasets, such as AIME25, and AMC. This makes me concerned about whether SIREN has truly expanded the reasoning boundary.
> > > > > >
> > > > > > But I am moved by the authors' hard work to address my questions, and would like to raise my score accordingly. I suggest the author do some hyperparameter search to find a better combination, which can acheive higher accuracies across different benchmarks.

---

> ### Author Response · Authors · 2025-12-03
> **Hyperparameter sensitivity analysis.**
>
> Thank you for raising this valuable concern regarding **the sensitivity of the top-P cutoff and the peak-entropy quantile**. We also appreciate your patience while we completed the sensitivity experiments, as the additional training required substantial computation time.
>
> In our main experiments, both hyperparameters were set to **0.8**. To evaluate whether our method is sensitive to these choices, we conducted an additional analysis by varying each parameter around 0.8 while keeping the other fixed. The results are summarized in the table below; in addition to maj and avg, we also report the full set of Pass@k metrics.
>
> |                                | AIME24   |          |          | AIME25   |          |          | AMC      |          |          | MATH500  |          |          | OlympiadBench |          |          | Avg.     |          |          |
> | ------------------------------ | -------- | -------- | -------- | -------- | -------- | -------- | -------- | -------- | -------- | -------- | -------- | -------- | ------------- | -------- | -------- | -------- | -------- | -------- |
> |                                | Pass     | Maj      | Avg      | Pass     | Maj      | Avg      | Pass     | Maj      | Avg      | Pass     | Maj      | Avg      | Pass          | Maj      | Avg      | Pass     | Maj      | Avg      |
> | TopP cutoff 0.8 + quantile 0.8 | 53.3     | **43.3** | **28.4** | 40.0     | **26.7** | **17.9** | **92.8** | 71.1     | 60.1     | **92.8** | 85.0     | 81.9     | 59.4          | 46.8     | 42.5     | 67.7     | **54.6** | **46.1** |
> |                                |          |          |          |          |          |          |          |          |          |          |          |          |               |          |          |          |          |          |
> | TopP cutoff 0.7                | **60.0** | 33.3     | 24.2     | 33.3     | 16.7     | 12.8     | 91.6     | 62.7     | 55.7     | 91.4     | 83.2     | 77.6     | 58.5          | 46.7     | 40.8     | 67.0     | 48.5     | 42.2     |
> | TopP cutoff 0.9                | 56.7     | 40.0     | 25.5     | 40.0     | 16.7     | 13.5     | 90.4     | 69.9     | 60.0     | 92.4     | 86.4     | 81.8     | 61.5          | 47.3     | 43.3     | 68.2     | 52.0     | 44.8     |
> |                                |          |          |          |          |          |          |          |          |          |          |          |          |               |          |          |          |          |          |
> | quantile 0.7                   | **60.0** | 33.3     | 21.6     | 40.0     | 23.3     | 15.5     | **92.8** | 71.1     | 61.6     | 90.8     | 84.6     | 80.1     | 56.3          | 42.1     | 38.7     | 68.0     | 50.9     | 43.5     |
> | quantile 0.75                  | **60.0** | **43.3** | 28.1     | 40.0     | 16.7     | 13.2     | 91.6     | **73.5** | **62.3** | **92.8** | **86.6** | **82.0** | **61.9**      | **50.4** | **44.2** | **69.3** | 54.1     | 46.0     |
> | quantile 0.85                  | 56.7     | 40.0     | 21.5     | **46.7** | 23.3     | 14.2     | **92.8** | 71.1     | 61.7     | 90.6     | 82.6     | 79.9     | 58.5          | 45.9     | 41.4     | 69.0     | 52.6     | 43.7     |
> | quantile 0.9                   | 46.7     | 26.7     | 16.0     | 40.0     | 23.3     | 13.1     | 84.3     | 69.9     | 54.9     | 92.0     | 84.4     | 78.3     | 61.2          | 47.1     | 41.1     | 64.8     | 50.3     | 40.7     |
>
> As shown in the results, tuning the topp cutoff and quantile within a wide range does not lead to a significant drop in pass@k—on the contrary, the average pass@k even slightly improves. Only when p=0.7 and quantile=0.9 is a decrease observed, which may be attributed to overly stringent conditions that exclude a large number of tokens from the entropy regularization calculation. Overall, adjusting p and quantile within a reasonable range does not cause severe performance degradation, indicating that SIREN exhibits strong robustness and does not rely heavily on precise hyperparameter tuning.

---

### Official Review · Reviewer_BzBv · 2025-10-22

**Soundness:** 2
**Presentation:** 2
**Contribution:** 2
**Rating:** 6
**Confidence:** 2

**Summary:**

This paper suggests a two-step entropy masking mechanism to enhance RLVR approaches. With this approach, semantically meaningless tokens disappeared, and the training process becomes stabilized.

**Strengths:**

- Main claim of the paper is backed by a wide range of experiments (math benchmarks) and ablation studies
- Implementation is provided in an anonymous repository.
- Concepts are clear: Top-p mask uses vocabulary subsets, and a peak-entropy mask selects them with quantile, and MSE is used for anchoring entropy

**Weaknesses:**

- The positioning of SIREN among entropy-regularized exploration methods was not sufficiently clear. I assume the extreme failure case in Fig.1 would not occur with Clip-Cov or Entropy Adv.
- Difficult to understand the flow of the recent approaches including Clip-Cov, Entropy Adv, ... (I have read Dr. GRPO, but wasn't aware of the details of the other baselines)

**Questions:**

- Figure 1 shows the side-effects of naive entropy regularization, which could be exaggerated. The extreme failure case illustrated in Figure 1 uses a naive approach with coefficient 0.005, but it's unclear whether more sophisticated baselines like Clip-Cov or Entropy Adv. also suffer from similar entropy explosions. Could you clarify if these recent methods exhibit comparable failure modes?
- The flow and positioning of recent entropy-based approaches is difficult to understand. While I've read about Dr.GRPO, the specific limitations of other baselines (Clip-Cov, Entropy Adv., RL on Forking Tokens) are not sufficiently explained. What are the concrete weaknesses of each method that SIREN addresses?
- It's challenging to fully grasp SIREN's positioning within the landscape of entropy-regularized exploration methods. The paper establishes that SIREN outperforms naive approaches, but the connection between SIREN's components (top-p mask, peak-entropy mask, self-anchored regularization) and the specific limitations of sophisticated baselines remains unclear. Could you provide explicit mappings, such as "top-p mask addresses Clip-Cov's limitation of X" or "peak-entropy mask improves upon Entropy Adv. by doing Y"?

---

> ### Author Response · Authors · 2025-11-21
> **Summary of Response**
>
> We sincerely thank Reviewer BzBv for the insightful comments. We address your comments as follows:
>
> - The flow of baseline methods.
> - SIREN's positioning within the baseline methods
>
> We hope our response addresses the reviewer’s concerns and clarifies our manuscript.

---

> ### Author Response · Authors · 2025-11-21
> **The flow of baseline methods**
>
> > Q:
> >
> > - It's unclear whether more sophisticated baselines like Clip-Cov or Entropy Adv. also suffer from similar entropy explosions.
> > - The flow and positioning of recent entropy-based approaches is difficult to understand. While I've read about Dr.GRPO, the specific limitations of other baselines (Clip-Cov, Entropy Adv., RL on Forking Tokens) are not sufficiently explained.
>
> We thank the reviewer for raising these points. While we briefly introduced the baselines in **Section 4.1 (Baseline methods)** and discussed their conceptual differences in Section 6 (**Related Work)**, we agree that the descriptions may not have been sufficiently detailed. We therefore provide a clearer explanation here, focusing on (i) how each method operates and (ii) why these baselines are generally not expected to exhibit the same type of entropy explosion that naïve entropy regularization suffers from.
>
> - **Clip-Cov / KL-Cov.**
>
> This work observes that changes in policy entropy are primarily driven by the covariance between action probabilities and logit updates, which is proportional to the advantage when using policy-gradient–style updates. High-probability actions with large advantages tend to decrease entropy, whereas rare actions with large advantages increase it. This work mitigates excessive entropy decrease by *suppressing updates* on tokens with high covariance: Clip-Cov samples a subset of these tokens and zeros out their policy loss, while KL-Cov selects the top-k% covariance tokens and applies an additional KL penalty to constrain their update magnitude. Importantly, these methods do not compute entropy or perform entropy-based regularization; instead, they indirectly regulate entropy by controlling updates on a small set of influential tokens. As a result, they are **not susceptible to entropy explosion**.
>
> - **Entropy Advantage.**
>
> This method applies an entropy-guided advantage shaping strategy to encourage exploratory reasoning. Concretely, it injects an entropy-dependent term $\psi(H_t) = \min(\alpha \cdot H_t^{\text{detach}}, |A_t|/\kappa)$ into the advantage function, forming $A^{\text{shaped}}_t = A_t + \psi(H_t)$
>
> Because entropy is *detached*, it does not introduce gradients that directly push entropy upward. Furthermore, the shaping term is strictly bounded by $|A_t|/\kappa$, preventing runaway growth. Thus, the method emphasizes high-entropy positions while **avoiding the uncontrolled positive feedback loop responsible for entropy explosion**.
>
> - **RL on Forking Tokens.**
>
> This work identifies that only a small fraction of tokens exhibit high entropy, and these “forking” tokens critically influence downstream reasoning paths. The method improves RLVR by restricting policy-gradient updates *only* to these high-entropy tokens, showing that using roughly 20% of the tokens is sufficient to match or exceed full-token optimization performance. Since the method does not apply entropy regularization at all and simply filters which tokens receive policy loss, it does **not introduce any mechanism that would drive entropy to diverge.**
>
> We have also added a dedicated subsection in the appendix providing detailed algorithmic descriptions of all baselines. We thank the reviewer again for the insightful comments and the opportunity to clarify these points more thoroughly.

---

> ### Author Response · Authors · 2025-11-21
> **SIREN's positioning within the baseline methods**
>
> > Q: It's challenging to fully grasp SIREN's positioning within the landscape of entropy-regularized exploration methods.
> >
>
> We appreciate the reviewer’s question regarding SIREN’s position within entropy-regularized exploration methods. Clarifying this indeed strengthens the paper.
>
> As discussed in the responses above, we provide a detailed comparison of the algorithmic flow of existing baselines. **These baselines do not explicitly regularize entropy.** Instead, their exploration behavior is controlled only indirectly—for example, by clipping specific tokens or detaching the entropy gradient—which shapes exploration heuristically rather than providing direct control over the entropy signal.
>
> In contrast, SIREN performs **explicit and stable entropy optimization,** with mechanisms that directly modulate and stabilize the entropy gradients. As shown in Figure 9, SIREN maintains both higher entropy (measured under the naive formulation) and more stable validation Pass@k throughout training. This empirically supports the distinction described above: SIREN enables more principled and reliable exploration dynamics than methods that rely solely on indirect heuristics.

---

### Official Review · Reviewer_ggMz · 2025-10-23

**Soundness:** 3
**Presentation:** 3
**Contribution:** 3
**Rating:** 6
**Confidence:** 4

**Summary:**

This paper analyzes the limitations of naive entropy regularization in RLVR, demonstrating that the vast action space and long trajectories in large reasoning models (LRMs) tend to flatten probability distributions across most positions. This observation highlights the necessity of controlling the effective scope of regularization. To address this, the authors propose SIREN which selects exploration scopes at both the action and trajectory levels for more effective entropy regularization, and transforms naive regularization into a self-anchored form to stabilize training.

**Strengths:**

1.  The paper revisits the reasons why traditional entropy regularization fails to improve performance for LLM reasoning, which is an important research direction.
2.  The authors provide clear visualizations that support their claims.
3.  The analysis convincingly attributes the failure of naive entropy regularization to the vast action space and long trajectories in LRMs.
4.  The experimental evaluation is comprehensive and includes up-to-date baselines.

**Weaknesses:**

1.  The proposed method is mostly based on heuristics; more theoretical analysis would strengthen the work.
2.  Regarding the Policy Nucleus: "Since these tokens consistently occupy the top ranks in the original model’s probability distribution, we adopt the terminology of Top-p sampling and refer to this subset as the nucleus." The authors seem to constrain exploration to tokens commonly seen in the base model. I wonder whether this might limit the exploration capability, i.e., some useful exploration could fall outside such a subset.

**Questions:**

See weakness.

---

> ### Author Response · Authors · 2025-11-21
> **Summary of Response**
>
> We sincerely thank Reviewer ggMz for the insightful comments. We address your comments as follows:
>
> - The theoretical foundation of SIREN.
> - Whether Top-p Masking restricts exploration.

---

> ### Author Response · Authors · 2025-11-21
> **The theoretical foundation of SIREN.**
>
> > Q:  The proposed method is mostly based on heuristics; more theoretical analysis would strengthen the work.
> >
>
> We thank the reviewer for raising this important point. To further strengthen the theoretical grounding of SIREN, we have added a detailed analysis in the appendix. Specifically, we show that naive entropy regularization in LRMs is either unstable, because it is prone to entropy explosion, or ineffective, because it provides negligible exploration incentives. In contrast, SIREN’s masking design explicitly bounds the entropy gradients, preventing unstable updates and ensuring that the entropy signal remains meaningful. Moreover, the masking mechanisms substantially reduce the variance of entropy gradients, which stabilizes optimization. The formal justification is provided in Appendix B.

---

> ### Author Response · Authors · 2025-11-21
> **Whether Top-p Masking restricts exploration.**
>
> > Q:  Regarding the Policy Nucleus: "Since these tokens consistently occupy the top ranks in the original model’s probability distribution, we adopt the terminology of Top-p sampling and refer to this subset as the nucleus." The authors seem to constrain exploration to tokens commonly seen in the base model. I wonder whether this might limit the exploration capability, i.e., some useful exploration could fall outside such a subset.
> >
>
> We understand the reviewer’s concern that the Top-p mask might limit exploration by excluding low-probability tokens. However, both theoretical analysis and our ablation study in Section 5.4 indicate that **the absence of Top-p Masking is strictly detrimental**.
>
> From a theoretical perspective, Top-p Masking significantly reduces the magnitude and variance of entropy gradients, preventing degenerate updates and raising the attainable performance ceiling of RLVR. Empirically, removing the Top-p Mask leads to a substantial performance drop. This shows that, rather than restricting exploration, Top-p Masking serves as a critical stabilizing component that **enables more effective exploration in practice**.

---

> ### Comment · Reviewer_ggMz · 2025-11-26
>
> Thanks for the authors' response; my review remains the same.

---

### Official Review · Reviewer_zyi9 · 2025-10-26

**Soundness:** 2
**Presentation:** 3
**Contribution:** 2
**Rating:** 2
**Confidence:** 4

**Summary:**

The paper explores to improve RLVR, particularly entropy collapse and premature convergence.
Authors argue that naive entropy regularization fails in LRMs due to their vast action spaces and long trajectories, often causing entropy explosions by flattening probability distributions across irrelevant tokens. The authors propose SIREN, a method that employs:
- a top-p mask to focus on semantically meaningful token subsets,
- a peak-entropy mask to target critical high-entropy positions in trajectories, and
- self-anchored regularization using mean squared error to stabilize entropy levels around an initial anchor.

Built on Dr.GRPO, SIREN demonstrates superior performance on benchmarks like AIME24/25, AMC, OlympiadBench, and MATH500.

**Strengths:**

- Pinpoints the problem of traditional naive entropy regularization
- Significant performance improvement compared to naive Dr.GRPO with entropy regularization.

**Weaknesses:**

- All three proposed mechanisms of SIREN: top-p mask, peak-entropy mask, self-anchored regulrization, does seem to be heuristics, being an ad-hoc solutions to seek for a structured exploration.
- Top-p mask and peak-entropy mask introduce additional hyper-parameters that need to be tuned. While the paper is motivated from the difficulty of tuning entropy coefficients, the difficulty of tuning newly introduced hyper-parameters are not discussed. The difficulty of tuning naive entropy regularizatin coefficient and self-anchored regularization coefficient is also not discussed (only results on 1~2 hyperparameters are shown), which I think is very important for the paper.
- The limitations of current algorithm - when the proposed algorithm would not work well - is not discussed. For example, there are datasets where SIREN gives large performance boost (AIME), and small hboost (MATH 500 or Olympiad). What makes the difference?
-  While the focus of the paper is mostly on (top-p mask, peak-entropy mask), most of the performance improvement comes from (self-anchored regulrization), which is not really a new idea. For example, SAC also uses entropy constraints rather than entropy regularization, which can be seem as a principle version of anchored regularization. I believe that entropy constraints should also be compared with self-anchored regularization.
- Shown results are somewhat indirect, and there seems not enough "direct" evidence that proposed SIREN mechanisms (top-p mask, peak-entropy mask) help to have more structured explorations. There is no other algorithms that achieves SIREN(0.005) level of entropy without entropy explosion, probably due to the absence of anchored regularization, and the performance improvements might be only caused by the anchored regularization. Since we are adding hyper-parameters, the small amount of performance improvement suggested would have been possible with any modifications when appropriately tuned.

**Questions:**

- In figure 6, it is said that naive regularization used coefficient of 0.001 (and seems to be learned well), where in the hyperparameter setting in appendix it says 0.0001 is used for naive reg and any higher value leads to instability. Which is true?

---

> ### Author Response · Authors · 2025-11-21
> **Summary of Response**
>
> We sincerely thank Reviewer zyi9 for the insightful comments. Your feedback has motivated us to provide additional analyses to clarify the motivation behind SIREN and to explain the potential reasons for its effectiveness. We address your comments as follows:
>
> - Motivation and effectiveness of SIREN.
> - Potential reasons for the varying improvements of SIREN on different datasets.
> - The improvements of SIREN are not solely due to self-anchored regularization; comparison with SAC-style constraints.
> - 'Direct' evidence of how SIREN promotes structured exploration.
> - Why baselines cannot reach SIREN’s entropy levels.
> - Clarification on the confusion arising from different hyperparameter settings.
>
> Regarding the newly introduced hyperparameters, **topP cutoff** and **entropy quantile**, we are currently running sensitivity analyses and will report the results during the rebuttal period. We hope that our responses address your concerns.

---

> > ### Comment · Reviewer_zyi9 · 2025-11-25
> >
> > While the rebuttals address some of my concerns, I will wait until the authors provide the sensitivity analyses results.

---

> ### Author Response · Authors · 2025-11-21
> **Motivation and effectiveness of SIREN.**
>
> > Q: All three proposed mechanisms of SIREN: top-p mask, peak-entropy mask, self-anchored regulrization, does seem to be heuristics, being an ad-hoc solutions to seek for a structured exploration.
> >
>
> We thank the reviewer for the question, and we address the effectiveness and contribution of SIREN from the following perspectives.
>
> 1. **Clear motivation rooted in a concrete failure mode.**
>
> We diagnose that naive entropy regularization in LRMs often triggers *entropy explosion*. Through analytical reasoning and empirical observations, we find that this failure mode arises from the **vast action space** and **long state trajectories** of LRMs: Actions with extremely small probabilities, which are often semantically irrelevant, are inadvertently encouraged, leading the action distribution to become overly uniform and diffuse throughout the sequence.
>
> To directly address this issue, we introduce **top-p masking** and **peak-entropy masking** to *bound* the effective entropy signal, and **self-anchored regularization** to *control its magnitude* and stabilize training.
>
> These components are designed as targeted solutions to the diagnosed failure mode; therefore, the overall method is **principled rather than arbitrary or ad-hoc**.
>
> 2. **Simple yet consistently effective.**
>
> Previous work, such as SimPO[1], s1[2], has been accepted despite using simple yet effective methods, demonstrating that simplicity alone does not preclude acceptance. Similarly, SIREN, although lightweight, directly constrains unstable entropy dynamics, yielding substantial improvements over prior entropy-based RL methods. Ablation studies further confirm that removing any single component significantly degrades performance, indicating that SIREN’s effectiveness stems from addressing the underlying structural issue rather than relying on superficial heuristics.
>
> [1] SimPO: Simple Preference Optimization with a Reference-Free Reward
>
> [2] s1: Simple test-time scaling

---

> > ### Comment · Reviewer_zyi9 · 2025-11-25
> >
> > 1. I respectively disagree with the authors on this point. I do not see any principles behind the proposed "direct treatments" to the observed failure modes.
> > 2. While I also voted marginal rejection for [1], I find this manuscript containing less empirical in-depth analyses compared to [1].

---

> > > ### Author Response · Authors · 2025-11-27
> > >
> > > > Q: I respectively disagree with the authors on this point. I do not see any principles behind the proposed "direct treatments" to the observed failure modes.
> > >
> > > ### The Behind Principles of Our Designs
> > >
> > > Thank you very much for raising this question. We hope that the newly added theoretical analysis in **Appendix B** can clarify the underlying principles behind our 'direct treatments' to the observed failure modes. Specifically, our analysis explains why naive entropy regularization tends to either *explode* or become *ineffective* from a gradient perspective.
> > >
> > > **Theoretical insight.**
> > >
> > > We decompose the action space into a high-probability *head* region and a near-zero-probability *tail* region. Our gradient analysis shows that the entropy gradients on tail actions can be extremely large, thereby dominating the optimization direction. With a **large** entropy coefficient, this tail-dominance leads to *explosion*; with a **small** coefficient, tail gradients are suppressed enough to avoid explosion, but the already small head gradients become negligible, resulting in *ineffectiveness*.
> > >
> > > Based on this observation, our method explicitly separates head and tail: we retain entropy gradients only for the head region through the **top-p mask**. Moreover, for highly deterministic tokens where the head becomes extremely small, our analysis shows that entropy-based exploration is no longer meaningful and the entropy gradients are negligible. This motivates the **peak-entropy mask**, which filters out such tokens.
> > >
> > > We further show that the masking mechanisms not only bound the magnitude of entropy gradients but also **reduce their variance**, which stabilizes optimization. The full derivation is provided in Appendix B.
> > >
> > >
> > > **Intuition behind the principle.**
> > >
> > > In short, our principle is to *control and reshape entropy gradients in a way that directly targets the causes of instability*, including tail-gradient explosion and head-gradient vanishing. The masks operate this principle in a simple and effective way.
> > >
> > > We hope this theoretical analysis helps clarify the principles behind our design. We also want to make sure we are correctly understanding your comment. **If the 'principle' you refer to means something different from what we addressed, we would appreciate it if you could briefly clarify it so that we can respond more precisely.**
> > >
> > > Thank you again for the valuable feedback.
> > >
> > > ---
> > > > Q: While I also voted marginal rejection for [1], I find this manuscript containing less empirical in-depth analyses compared to [1].
> > >
> > > ### Empirical In-depth Analysis
> > >
> > > We cite [1] only to illustrate that a simple yet effective paradigm can still carry meaningful value. We would also like to clarify the **empirical in-depth analysis** we conducted. We perform both **qualitative** and **quantitative** experiments to demonstrate the effectiveness of our method.
> > >
> > > For the **qualitative analysis**, we examine SIREN’s outputs in **Appendix F.2**, where we provide finer-grained stepwise examples that use more tokens and include complete verification. We also present **word clouds** in **Appendix F.1** that visualize the tokens selected by the mask. These tokens are core terms closely related to mathematical problem statements and reasoning, such as *given*, *the*, *to*, *answer*, and *consider*, along with mathematical concepts such as *product*, *sum*, *value,* and *equation*.
> > >
> > > For the **quantitative analysis**, Section 5 reports several results. We evaluate the **pass@k performance** of SIREN in **Figure 4**, which shows its effectiveness in expanding the reasoning boundary. We compute **response perplexity** in **Figure 5**, showing that SIREN preserves greater diversity than naive methods, which facilitates exploration. We also analyze **training dynamics**, including entropy variations and validation pass@16 in **Figure 6**, which further confirm SIREN’s ability to control exploration during training.
> > >
> > > Given these qualitative and quantitative results, we believe the reviewer’s concern about the lack of in-depth empirical analysis has been adequately addressed. If there are specific additional analyses you consider important, we would be grateful if you could point them out, and we would be happy to include them.

---

> ### Author Response · Authors · 2025-11-21
> **Potential reasons for the varying improvements of SIREN on different datasets.**
>
> > Q: The limitations of current algorithm - when the proposed algorithm would not work well - is not discussed. For example, there are datasets where SIREN gives large performance boost (AIME), and small boost (MATH 500 or Olympiad). What makes the difference?
> >
>
> SIREN’s improvement varies across datasets for reasons that go beyond the algorithm itself. The gain is influenced by **(i) the size of the solution space**, **(ii) the inherent difficulty and diversity of the tasks**, and **(iii) the model’s prior proficiency on that dataset before re-ranking**.
>
> Datasets like AIME have a highly regular structure and a small, constrained answer space, which allows SIREN to leverage structural signals more effectively and thus achieve larger improvements. In contrast, MATH500 and OlympiadBench have much larger and more diverse solution spaces, and the base model already performs relatively strongly on these tasks. Under such conditions, the potential headroom for re-ranking is naturally smaller. This may explain why the observed gains differ across datasets, and reflects differences in **data properties**, not limitations of the SIREN algorithm.
>
> It is also worth noting that **other baselines that explicitly aim to maintain entropy levels**, such as **KL-cov** and **Clip-cov**[3], exhibit a very similar pattern to SIREN. Compared with *clip-higher* (which corresponds to our baseline, as both use higher clip ratio), these entropy-preserving variants also show **much larger improvements on AIME24/25 than on the other datasets**.
>
> [3] The Entropy Mechanism of Reinforcement Learning for Reasoning Language Models.

---

> ### Author Response · Authors · 2025-11-21
> **The improvements of SIREN are not solely due to self-anchored regularization; comparison with SAC-style constraints.**
>
> > Q: While the focus of the paper is mostly on (top-p mask, peak-entropy mask), most of the performance improvement comes from (self-anchored regularization), which is not really a new idea. For example, .a SAC also uses entropy constraints rather than entropy regularization, which can be seem as a principle version of anchored regularization. I believe that entropy constraints should also be compared with self-anchored regularization.
> >
>
> We appreciate the reviewer’s insightful comments. We would like to clarify two points.
>
> 1. **Self-anchored regularization is not the sole source of improvement.**
>
> In Sec. 5 Table 2, we provide an ablation using *only* self-anchored regularization, which yields **52.3 avg-maj** and **43.6 avg-avg**, both significantly below the full SIREN performance (**54.6 / 46.1**). This demonstrates that the masking components contribute substantial gains. Moreover, as analyzed in Appendix B, the masking mechanisms provide complementary benefits: they **bound entropy gradients**, reducing both excessive and insufficient exploration, and **decrease the variance** of entropy gradients, which stabilizes the optimization direction and makes it less likely to get stuck in local optima. Therefore, the observed improvement is not solely attributable to self-anchoring, nor is self-anchoring intended to serve as the main module.
>
> **(2) Relationship to SAC and why self-anchored works better in LRM.**
>
> We partially agree with the reviewer that self-anchored regularization aligns with the same principle as SAC’s dual-gradient entropy constraint: both aim to keep the policy entropy close to a target value. In response, we have added a direct comparison experiment between SIREN’s self-anchored regularization and a SAC-style dual-gradient update.
>
> We observe that dual-gradient regularization yields clear improvements over naive regularization, but **it still underperforms SIREN**. While SAC-style methods can make entropy more controllable by adapting the entropy coefficient online, they introduce additional tuning complexity: the initial entropy coefficient and its learning-rate require careful, often brittle, adjustment. In contrast, SIREN only requires driving policy entropy toward a target value, which substantially reduces the number of tunable hyperparameters and the sensitivity of training to their settings.
>
> |  | AIME24 |  | AIME25 |  | AMC |  | MATH500 |  | OlympiadBench |  | Average |  |
> | --- | --- | --- | --- | --- | --- | --- | --- | --- | --- | --- | --- | --- |
> |  | Maj@32 | Avg@32 | Maj@32 | Avg@32 | Maj@32 | Avg@32 | Maj@8 | Avg@8 | Maj@8 | Avg@8 | Maj@k | Avg@k |
> | Dual Gradient Reg. | 36.7 | **30.2** | 20.0 | 16.7 | 68.7 | **63.7** | 82.8 | 79.6 | 43.7 | 40.7 | 50.4 | **46.2** |
> | SIREN | **43.3** | 28.4 | **26.7** | **17.9** | **71.1** | 60.1 | **85.0** | **81.9** | **46.8** | **42.5** | **54.6** | 46.1 |

---

> ### Author Response · Authors · 2025-11-21
> **'Direct' evidence of how SIREN promotes structured exploration.**
>
> > Q: Shown results are somewhat indirect, and there seems not enough "direct" evidence that proposed SIREN mechanisms (top-p mask, peak-entropy mask) help to have more structured explorations.
> >
>
> We provide a qualitative analysis of SIREN’s responses in Appendix G. Our observations indicate that SIREN tends to break down problems into finer-grained steps using more tokens and includes a complete verification process, which results in more accurate answers. We believe this can serve as **direct evidence** that the SIREN mechanisms facilitate more structured exploration. In addition to the qualitative analysis, Section 5 demonstrates that SIREN **improves pass@k** across different datasets, thereby **extending the potential reasoning boundaries**. Moreover, it maintains **higher perplexity** compared to the DR.GRPO baseline, effectively mitigating the overly convergent behavior caused by RLVR, as observed in prior work ([4]).
>
> [4] Does Reinforcement Learning Really Incentivize Reasoning Capacity in LLMs Beyond the Base Model?

---

> ### Author Response · Authors · 2025-11-21
> **Why baselines cannot reach SIREN’s entropy levels.**
>
> > Q: There is no other algorithms that achieves SIREN(0.005) level of entropy without entropy explosion, probably due to the absence of anchored regularization, and the performance improvements might be only caused by the anchored regularization. [Q5].b Since we are adding hyper-parameters, the small amount of performance improvement suggested would have been possible with any modifications when appropriately tuned.
> >
>
> We appreciate the reviewer’s comment. We would like to emphasize the difference between SIREN and existing baselines. Methods such as *entropy-adv[5]* or *on-forking-tokens[6]* **do not use entropy regularization;** instead, they detach the entropy gradient and only use the entropy value to **assign credit.** As a result, these methods cannot explicitly increase entropy. In fact, in simple datasets, rewarding high-entropy correct tokens may even reinforce the current answers, **which can further accelerate entropy convergence**. Anchored regularization, as an explicit mechanism to control entropy, **cannot be incorporated into these baselines**.
>
> In addition, in Section 5.4, we present an ablation study using only self-anchored regularization (detailed results in Table 3). While the average improvements may appear small, this is largely due to aggregation over all instances. Looking at the raw data, s**elf-anchored regularization alone performs worse on more challenging datasets**—for example, on AIME24, Maj@K is 13.3 points lower than SIREN. This indicates that the improvements are not merely the result of hyperparameter tuning, but reflect the effectiveness of the full SIREN mechanisms.
>
> [5] Reasoning with Exploration: An Entropy Perspective.
>
> [6] Beyond the 80/20 Rule: High-Entropy Minority Tokens Drive Effective Reinforcement Learning for LLM Reasoning

---

> ### Author Response · Authors · 2025-11-21
> **Clarification on the confusion arising from different hyperparameter settings.**
>
> > Q: In figure 6, it is said that naive regularization used coefficient of 0.001 (and seems to be learned well), where in the hyperparameter setting in appendix it says 0.0001 is used for naive reg and any higher value leads to instability. Which is true?
> >
>
> The coefficient shown in Fig. 6 (0.001) corresponds to the setting used in the standalone naive-entropy baseline, where no SIREN masking components are applied. In contrast, the value 0.0001 refers to **the ablation setting in which self-anchored regularization is removed** but SIREN’s masking mechanisms are retained. The difference in these coefficients arises from the way entropy loss is computed. In the naive baseline, the cumulative entropy is calculated in the same manner as the policy loss and is divided by MAX_LENGTH. In SIREN, however, we compute the mean over the effective entropy directly, which results in a larger raw value. Consequently, a smaller $\alpha$ is required for SIREN.

---

> ### Author Response · Authors · 2025-12-03
> **Hyperparameter sensitivity analysis.**
>
> Thank you for raising this valuable concern regarding **the sensitivity of the top-P cutoff and the peak-entropy quantile**. We also appreciate your patience while we completed the sensitivity experiments, as the additional training required substantial computation time.
>
> In our main experiments, both hyperparameters were set to **0.8**. To evaluate whether our method is sensitive to these choices, we conducted an additional analysis by varying each parameter around 0.8 while keeping the other fixed.
>
> |                                | AIME24   |          | AIME25   |          | AMC      |          | MATH500  |          | OlympiadBench |          | Avg.     |          |
> | ------------------------------ | -------- | -------- | -------- | -------- | -------- | -------- | -------- | -------- | ------------- | -------- | -------- | -------- |
> |                                | Maj      | Avg      | Maj      | Avg      | Maj      | Avg      | Maj      | Avg      | Maj           | Avg      | Maj      | Avg      |
> | TopP cutoff 0.8 + quantile 0.8 | **43.3** | **28.4** | **26.7** | **17.9** | 71.1     | 60.1     | 85.0     | 81.9     | 46.8          | 42.5     | **54.6** | **46.1** |
> |                                |          |          |          |          |          |          |          |          |               |          |          |          |
> | TopP cutoff 0.7                | 33.3     | 24.2     | 16.7     | 12.8     | 62.7     | 55.7     | 83.2     | 77.6     | 46.7          | 40.8     | 48.5     | 42.2     |
> | TopP cutoff 0.9                | 40.0     | 25.5     | 16.7     | 13.5     | 69.9     | 60.0     | 86.4     | 81.8     | 47.3          | 43.3     | 52.0     | 44.8     |
> |                                |          |          |          |          |          |          |          |          |               |          |          |          |
> | Quantile 0.7                   | 33.3     | 21.6     | 23.3     | 15.5     | 71.1     | 61.6     | 84.6     | 80.1     | 42.1          | 38.7     | 50.9     | 43.5     |
> | Quantile 0.75                  | **43.3** | 28.1     | 16.7     | 13.2     | **73.5** | **62.3** | **86.6** | **82.0** | **50.4**      | **44.2** | 54.1     | 46.0     |
> | Quantile 0.85                  | 40.0     | 21.5     | 23.3     | 14.2     | 71.1     | 61.7     | 82.6     | 79.9     | 45.9          | 41.4     | 52.6     | 43.7     |
> | Quantile 0.9                   | 26.7     | 16.0     | 23.3     | 13.1     | 69.9     | 54.9     | 84.4     | 78.3     | 47.1          | 41.1     | 50.3     | 40.7     |
>
> As shown, increasing the top-p cutoff value leads to only a slight degradation in model performance. However, when the value is reduced to 0.7, performance drops significantly. This is likely because a cutoff of 0.7 is too small in terms of probability; many tokens end up with zero entropy after applying the top-p mask, causing them to be filtered out. With too few tokens left for entropy regularization to take effect, the model’s performance suffers.
>
> Regarding quantile, adjusting it by ±0.05 results in stable performance, and even a reduction of 0.1 maintains acceptable avg@k. In contrast, increasing the quantile by 0.1 leads to a more substantial performance decline. This may be attributed to excessively large quantiles filtering out effective logical tokens. Overall, adjusting p and quantile within a reasonable range does not cause severe performance degradation, indicating that SIREN is robust and does not rely heavily on precise hyperparameter tuning.
>
> We have included this table in the revised paper to improve the clarity and completeness of our analysis.

---

### Comment · Area_Chair_a1aY · 2025-11-25

Dear Reviewers

Thank you for your time and help for reviews.
The author-reviewer discussion due is in one week. If you have not done yet, please review the authors' rebuttal for the paper under your evaluation and engage in discussion with authors.

Thank you again.
Best,

Area Chair

---

### Author Response · Authors · 2025-12-03
**Overall Summary of Key Revisions and Clarifications**

We sincerely thank the reviewers and the area chair for the time and effort devoted to evaluating our submission. The constructive feedback we received has been instrumental in refining and strengthening our work. Below we provide a concise summary of our rebuttal to help the area chair efficiently assess the current status of the paper. Our responses mainly address the following key aspects, with the corresponding revisions in the paper highlighted in blue:

- **Clear and Sufficient Motivation**
We clarified the core motivation of our method [Refer to Reviewer_zyi9.W1] and highlighted its conceptual distinctions from existing baselines [Refer to Reviewer_BzBv.W1, Reviewer_BzBv.Q3]. We also included a more detailed description of the baseline methods, covering their algorithmic flow [Refer to Reviewer_BzBv.W2] and explaining why other baselines do not cause entropy explosion [Refer to W3.Q1], to help reviewers better understand the position of our method [Refer to Reviewer_BzBv.W1, Reviewer_BzBv.Q3]. Please see the updated content in Appendix C.2.
- **Strengthened Theoretical Grounding**
We expanded and clarified the theoretical underpinnings of our approach, including the theoretical explanation for why naïve entropy regularization fails [Refer to Reviewer_ys5j.W3], as well as the theoretical validity of SIREN, specifically its ability to bound entropy gradients and reduce gradient variance [Refer to Reviewer_zyi9.W1, Reviewer_ggMz.W1, Reviewer_ys5j.W2].  Detailed elaborations are provided in Appendix B.
- **Additional Experimental Evidence**
  We conducted new experiments directly addressing reviewers’ concerns, including hyperparameter sensitivity analysis [Refer to Reviewer_zyi9.W2, Reviewer_ys5j.Q1; see Table 3 and Table 6], additional comparisons with new baselines [Refer to Reviewer_zyi9.W4, Reviewer_ys5j.Q3; see table 1], and evaluation using Pass@k metrics [Refer to the additional comment from Reviewer_ys5j]. All new results consistently support the robustness and advantages of our method.
- **More Comprehensive Quantitative Analysis**
  We added detailed quantitative evaluations to provide more thorough and transparent evidence of SIREN’s effectiveness [Refer to Reviewer_zyi9.W5; see Appendix G]. These analyses more intuitively demonstrate that SIREN achieves superior performance by refining the reasoning process and offering a more detailed verification procedure.

Again, we would like to emphasize that our paper delivers three core contributions:

- **Analysis of the limitations of naive entropy regularization in RLVR.** We show that the vast action space and long trajectories flatten probability distributions across most positions. We also provide a theoretical explanation for this failure.
- **Proposal of SIREN.** We introduce a novel method that selects exploration scopes at both the action and trajectory levels for effective entropy regularization, while transforming naive regularization into a self-anchored form to stabilize training. SIREN theoretically addresses the problem by bounding entropy gradients and reducing gradient variance, leading to more stable and effective training.
- **Empirical performance.** We achieve competitive results on mathematical benchmarks, showing consistent improvements over other entropy-related baselines on maj@k and avg@k. Additional hyperparameter sensitivity analyses demonstrate that SIREN is robust to parameter changes.

Below, we also provide a rebuttal timeline and a detailed summary that distills each reviewer’s concerns and our corresponding rebuttal strategies into a concise and easy-to-assess format to assist the new area chair in their evaluation.

---

> ### Author Response · Authors · 2025-12-03
> **Timeline of Rebuttal Updates**
>
> We submitted all initial rebuttal responses on Nov 21 (AoE), except for the hyperparameter sensitivity analysis experiment, which was supplemented on Dec 2 (AoE) due to time and resource constraints.
>
> During the rebuttal period, three reviewers provided feedback:
>
> - Nov 21 (AoE): Reviewer ys5j began engaging actively in discussions with us.
> - Nov 24 (AoE): Reviewer zyi9 stated that we "addressed some of my concerns" and awaited the hyperparameter sensitivity analysis results.
> - Nov 25 (AoE): Reviewer ggMz maintained their original comments unchanged.
> - Nov 27 (AoE): Reviewer ys5j raised the score to 6.

---

> ### Author Response · Authors · 2025-12-03
> **Detailed Summary of Individual Reviewer Concerns and Our Responses**
>
> We thank the reviewers, originally assigned AC, and newly assigned AC for all their efforts during this review process. We genuinely believe that the review process helped us improve the paper. In order to facilitate the new AC's understanding of the rebuttal status, we provide this comment to summarize each reviewer's questions and our response strategies individually.
>
> ## Reviewer zyi9 — Score 2
>
> *The Reviewer zyi9 raised concerns as follows:*
>
> > Principle behind the proposed “direct treatments” for failure modes.
>
> We added theoretical insight explaining the design of SIREN and clarified why the method directly addresses the observed failure patterns.
>
> > Additional hyperparameters introduced by top-p and peak-entropy masks.
>
> We conducted hyperparameter sensitivity experiments and found that SIREN remains robust across a wide range of settings; see the revisions in Tables 3 and 6.
>
> > Lack of discussion on limitations of the current algorithm.
>
> We explained that the varying performance gains arise from task-dependent factors, and noted that baselines like KL-Cov and Clip-Cov exhibit the same pattern, suggesting this variability is not specific to SIREN.
>
> > Comparison with other entropy-constraint baselines.
>
> We added comparisons with SAC-style dual-gradient regularization and found that SIREN performs on par with or better than the baseline, with the relevant revisions presented in Table 1.
>
> > Lack of direct evidence that top-p and peak-entropy masks lead to more structured exploration.
>
> We added annotated qualitative analyses of SIREN’s responses (Appendix G), providing direct evidence of improved structured exploration.
>
> > Concern that SIREN’s advantage may stem only from anchored regularization, and whether anchored regularization alone could improve baseline performance.
>
> Our theoretical analysis and ablation results show that SIREN’s effectiveness is not attributable to this component alone. We further clarified that the baselines do not include any form of anchored regularization, and due to their algorithmic structure, this mechanism cannot be trivially integrated into them.
>
> > Inconsistency in entropy-coefficient settings (0.001 vs. 0.0001).
>
> We clarified that one setting corresponds to a baseline experiment and the other to an ablation study. They belong to different experimental designs and are not contradictory.
>
> **Reply:**
>
> The reviewer responded on Nov 24(AoE), noting that the rebuttal addressed some of their concerns.
>
> ---
>
> ## Reviewer ggMz - Score 6
>
> *The Reviewer ggMz raised concerns as follows:*
>
> > Request for deeper theoretical analysis.
>
> We added a detailed theoretical analysis in Appendix B.
>
> > Concern that top-p masking might limit exploration.
>
> We acknowledged this possibility but explained that extremely low-probability, semantically irrelevant tokens rarely lead toward correct solutions. Both our theoretical analysis and ablation studies confirm that top-p masking provides a net benefit.
>
> **Reply:**
>
> The reviewer responded that their comments remain unchanged.
>
> ---
>
> ## Reviewer BzBv - Score 6
>
> *The Reviewer BzBv raised concerns as follows:*
>
> > Request for clearer flow of recent approaches such as Clip-Cov and Entropy-Adv.
>
> We provided detailed algorithmic procedures for all baselines (revision in Appendix C.2), highlighting how each interacts with entropy.
>
> > Clarification of SIREN’s position among entropy-regularized exploration methods.
>
> We emphasized that SIREN directly controls the entropy level, whereas prior methods rely on shaping advantages or indirectly influencing entropy through auxiliary mechanisms.
>
> ---
> ## Reviewer ys5j - Score 2 => Score 6
>
> *The Reviewer ys5j raised concerns as follows:*
>
> > Distinction and novelty.
>
> We clarified that our novelty lies in diagnosing the failure of naïve entropy regularization, analyzing its underlying causes, and proposing direct mechanisms to resolve them. We explicitly contrasted our method with prior studies.
>
> > Lack of theoretical grounding.
>
> We added a detailed theoretical analysis in Appendix B.
>
> > Concern about over-claiming (“rethinking entropy regularization”, “foundational insight”).
>
> We clarified that our analysis uses both sequence-level and token-level visualizations to reveal new observations, and we additionally incorporated theoretical explanations of the failure phenomenon as **a new analytical tool.**
>
> We also carefully reconsidered the framing and proposed an alternative, more grounded title.
>
> > Sensitivity to p (top-p) and τ (peak-entropy quantile).
>
> We included sensitivity analyses for both hyperparameters in Table 3 and Table 6.
>
> > Comparison with adaptive-entropy methods (e.g., SAC-style).
>
> We added direct comparisons with SAC-style regularization in Table 1.
>
> > Additional Pass@k results.
>
> We already reported Pass@k in Section 5.2 and now highlight the results more explicitly.
>
> **Reply:**
>
> The reviewer ys5j replied on Nov 21 and raised the score to 6 on Nov 27 (AoE).

---

### Meta-Review · Area_Chair_hATe · 2026-01-09

**Summary:**

The reviewers initially recognized the significance of addressing entropy collapse in Reinforcement Learning with Verifiable Rewards (RLVR) but raised substantial concerns regarding the perceived heuristic nature of the proposed SIREN method, particularly the top-p and peak-entropy masking mechanisms. The primary reservations focused on the lack of a rigorous theoretical foundation explaining why these specific masks effectively mitigate entropy explosion compared to naive regularization, the introduction of sensitive hyperparameters, and the ambiguity of the method's positioning relative to concurrent baselines like Clip-Cov and Entropy-Adv. Additionally, reviewers questioned whether the performance gains were driven solely by the self-anchored regularization component rather than the novel masking techniques and requested more comprehensive empirical evidence, such as Pass@k metrics and sensitivity analyses, to validate the claims of improved exploration and reasoning boundaries.

**Reviewer Concerns:**

The rebuttal successfully addressed the majority of the empirical and comparative concerns, as the authors provided a new theoretical analysis in Appendix B to justify the gradient dynamics of the masking, clarified the algorithmic distinctions from baselines, and added the requested hyperparameter sensitivity studies and Pass@k evaluations which notably satisfied Reviewer ys5j. However, outstanding concerns remain for Reviewer zyi9, who continued to view the proposed "direct treatments" as ad-hoc heuristics rather than principled solutions despite the added theoretical context, and Reviewer ggMz retained a reservation that the top-p masking strategy might inherently limit the exploration of potentially useful low-probability tokens, suggesting that while the method works empirically, the theoretical justification for its design choices remains a point of friction for some evaluators.

**Reviewer Scores:**

Reviewer ys5j explicitly raised their score from 2 to 6 during the discussion period after the authors provided the requested Pass@k data and sensitivity analysis, indicating their concerns were fully resolved. Reviewers ggMz and BzBv maintained their scores of 6, signaling that the clarifications regarding baselines and theoretical grounding were sufficient to sustain their support for acceptance. Reviewer zyi9 maintained a score of 2; however, since they acknowledged that the sensitivity analysis addressed their specific empirical requests, it is plausible that with further engagement regarding the new theoretical appendix, they might have raised their score to a borderline 4, though their fundamental objection to the heuristic nature of the method suggests they would likely remain the most critical voice.

---

### Decision · Program_Chairs · 2026-01-26

Reject